

# Improved Simulations of Biomass Burning Aerosol Optical Properties and Lifetimes in the NASA GEOS Model during the ORACLES-I Campaign

Sampa Das[1,2] Peter R. Colarco[1], Huisheng Bian[1,3], and Santiago Gassó[2,4]

[1]Atmospheric Chemistry and Dynamics Laboratory, NASA Goddard Space Flight Center, Greenbelt, Maryland, USA
[2]Earth System Science Interdisciplinary Center, University of Maryland, College Park, Maryland, USA
[3]Goddard Earth Sciences Technology and Research II, University of Maryland, Baltimore County, Maryland, USA
[4]Climate and Radiation Laboratory, NASA Goddard Space Flight Center, Greenbelt, Maryland, USA

*Correspondence to*: Sampa Das (sampa.das@nasa.gov) and Peter Colarco (peter.r.colarco@nasa.gov)

**Abstract**. In order to improve aerosol representation in the NASA Goddard Earth Observing System (GEOS) model, we evaluated simulations of the aerosol properties and transport over the southern African biomass burning source and outflow region using observations made during the first deployment of the ORACLES (ObseRvations of Aerosols above CLouds and their intEractionS) field campaign in September 2016. An example case study of September 24 was analyzed in detail, during which aircraft-based in-situ and remote sensing observations showed the presence of a
multi-layered smoke plume structure with significant vertical variation in single scattering albedo (SSA). Our baseline GEOS simulations were not able to represent the observed SSA variation, nor the observed organic aerosol-to-black carbon ratio (OA:BC). Analyzing the simulated smoke age suggests that the higher altitude, less absorbing smoke plume was younger (~4 days), while the lower altitude and more absorbing smoke plume was older (~7 days). We hypothesize a chemical or microphysical loss process exists to explain the change in aerosol absorption as the smoke
plume ages, and we apply a simple 6-day e-folding loss rate to the model hydrophilic biomass burning OA to mimic this process. Adding this loss process required some adjustment to the model assumed scaling factors of aerosol emissions to conserve the regional aerosol loading and further improve the simulated OA:BC ratio. Accordingly, we have increased our biomass burning emissions of OA by 60% and biomass burning BC by 15%. We also utilized the ORACLES airborne observations to better constrain the simulation of aerosol optical properties, adjusting the assumed
particle size, hygroscopic growth, and absorption. Our final GEOS model simulation with additional OA loss and updated optics showed a better performance in simulating AOD and SSA compared to independent ground and space-based retrievals for the entire month of September 2016, including the OMI Aerosol Index. In terms of radiative implications of our model adjustments, the final GEOS simulation suggested a decreased atmospheric warming of about 10% (~2 W m$^{-2}$) over the south-east Atlantic region and above the stratocumulus cloud decks compared to the
model baseline simulations. These results improve the representation of the smoke age, transport, and optical properties in Earth system models.

## 1    Introduction

Smoke plumes emitted by biomass burning in the southern African region are transported over the southeast (SE) Atlantic during the peak fire season (August-October) every year, and they modify the regional energy budget
via direct, indirect, and semi-direct effects of aerosols contained in these plumes (Lu et al., 2018; Gordon et al., 2018; Das et al., 2020). Observation-based estimates of the direct radiative effects (DRE) of aerosol over the SE Atlantic showed that at the top of the atmosphere (TOA) the DRE of biomass burning aerosols for a given set of aerosol optical properties can be positive (that is, radiatively warming) or negative (radiatively cooling), depending on the albedo and coverage of the underlying clouds (Chand et al., 2009; Zhang et al., 2016; Wilcox, 2012; Keil and Haywood, 2003).
However, the magnitudes of these observation-based aerosol DRE estimates remain uncertain partly due to the poorly quantified aerosol radiative properties of aerosols above clouds (Kacenelenbogen et al., 2019). For the modeling-based studies as well, there is no consensus on the sign or magnitude of the direct aerosol forcing over this region (Myhre et al., 2013; Schulz et al., 2006). In addition, the aerosol DRE estimated from satellites typically exceed model estimates over this region (de Graaf et al., 2020). One of the possible reasons of the DRE mismatch could be that the
aerosol optical properties assumptions, especially single-scattering albedo (SSA)—which determines aerosol absorption—has large variability among models, and model SSAs were found to be usually higher than the aircraft measurements (Doherty et al., 2022; Shinozuka et al., 2020; Mallet et al., 2021).





Climate models usually derive their aerosol optical properties (extinction coefficient, SSA and phase function) using a database of Mie theory-based calculations (e.g., Optical Properties of Aerosols and Clouds or OPAC (Hess et al., 1998)) available for a range of wavelengths and relative humidity (RH) values, under the assumptions of aerosol mixing state, microphysical properties (size distribution, refractive index, etc.) and water uptake (or hygroscopic growth factor). In general, these properties (or their functional forms) are prescribed in the model, and especially in models that lack complex aerosol microphysical and chemical representations, these properties represent a hard constraint on the possible parameter space in the model. Furthermore, aerosol properties set in the model near biomass burning sources may not be representative of the changes in aerosol properties during long-range transport of smoke plumes due to changing composition and size, photochemistry, and humidification of aerosol particles (Zhao et al., 2015; Wong et al., 2017; Freney et al., 2010; Cubison et al., 2011).

In this study we investigate the properties of biomass burning aerosols over the SE Atlantic as simulated in the NASA Goddard Earth Observing System (GEOS) model and observed by airborne, space-based, and ground-based measurements taken during the ObseRvations of Aerosols above CLouds and their intEractionS (ORACLES) field campaign (Redemann et al., 2021). Recent advances in the GEOS model have improved the representation of aerosol properties, and especially the representation of carbonaceous aerosols (Colarco et al., 2017; see details in section 2.4). We focus in this paper on the results of GEOS simulations made over the period of the ORACLES-I field campaign, which was conducted out of Walvis Bay, Namibia, during late August-September 2016. The ORACLES-I deployment included two aircraft: the NASA P-3 for full atmospheric profiling and low-to-mid-level in-situ sampling, and the high-altitude NASA ER-2 for remote sensing observations (Fig. 1a). The two subsequent phases of ORACLES (August 2017 and October 2018) only had the P-3 aircraft, so the focus on ORACLES-I gives the most data to evaluate our model simulations.

Our specific objective in this study is to understand and improve the simulated aerosol optical properties in the GEOS model in the context of the ORACLES observations. Significantly, changes in SSA along the aerosol vertical profiles and possibly with aging smoke plumes were observed during ORACLES-I for several flights (Dobracki et al., 2023; Redemann et al., 2021). Here, we utilize the GEOS model to investigate further this issue of SSA variability and present a simplistic approach to capture the observed SSA change within aging smoke plumes for chemistry transport models. Once we tuned the model aerosol optics and lifetimes based on ORACLES aircraft observations, we also utilize the larger spatial coverage of the satellite and ground-based observations to evaluate the model aerosol properties over the complete southern African biomass burning source and outflow regions.

The paper is organized as follows: Section 2 provides a brief description of the observations used in this study along with the description of the model set-up and various capabilities of GEOS that we utilize to complement our analysis. Section 3 discusses the model-data comparisons and the radiative implications of tuning the aerosol properties. Finally, Section 4 summarizes the major findings of our study and discusses future directions for subsequent modeling studies in this regard.

## 2 Approach and Methods

### 2.1 ORACLES 2016 Aircraft-based Observations

We provide here brief details of the instruments relevant to our study based on the particle properties measured. Further details of the ORACLES-I instrumentation and related uncertainties are provided in Shinozuka et al. (2020), Redemann et al. (2021), and references therein.

**Aerosol Optical Properties**: The Hawaii Group for Environmental Aerosol Research (HiGEAR) operated several in-situ instruments on the P-3. These include two Radiance Research particle soot absorption photometers (PSAPs) to measure the aerosol absorption coefficients (at 470, 530, and 660 nm), and two TSI (model 3563) nephelometers (at 450, 550, and 700 nm) to measure the aerosol light scattering coefficients. In addition to the TSI nephelometers, two single wavelength nephelometers (at 540 nm, Radiance Research, M903) were operated concurrently to measure the increase in light scattering as function of RH. The humidified nephelometer was operated close to 80% RH while the dry unit was maintained below 40% (Howell et al., 2006; Pistone et al., 2019). For in-situ single scatter albedo (SSA) or extinction calculation, the measured PSAP absorption was interpolated to nephelometer wavelengths before combining with its scattering data. For both SSA and extinction, we use the corrected or processed data (called SSA_ATP and Exttotal_ATP) that are provided at a time resolution of 1 second within the "merged" files at the ORACLES ESPO Data Archive: https://espo.nasa.gov/oracles/archive/browse/oracles/id8, last accessed: 22 March, 2022. Here, PSAP absorption corrections were performed according to an updated algorithm (Virkkula, 2010) and nephelometer measured scattering were corrected according to Anderson and Ogren, (1998). The in-situ SSA and



extinctions are measured and reported at low (<40%) RH. Additionally, we only use the measurements for which mid-visible dry extinctions are greater than 10 Mm⁻¹ (Pistone et al. 2019).

In addition to dry in-situ SSA, we also use the retrievals of column-integrated SSA for ambient conditions from the Spectrometer for Sky-Scanning Sun-Tracking Atmospheric Research (4STAR) instrument, which was also on-board the P-3. 4STAR is an airborne hyperspectral (350-1700 nm) sun photometer which can make direct-beam
measurements (sun-tracking mode) for retrievals of column aerosol optical depth (AOD) above flight level (LeBlanc et al., 2020). Under certain level-flying conditions, 4STAR can also perform AERONET-like sky scans in either the principal-plane or almucantar (sky scanning mode). The 4STAR sky scans were processed using a modified version of the Version 2 AERONET (Aerosol Robotic Network) retrieval algorithm described in Dubovik and King, (2000). However, due to suspected stray light contamination within the 4STAR spectrometer around 440 nm, the set of input
wavelengths for 4STAR were modified to be 400, 500, 675, 870, and 995 nm (as opposed to the standard AERONET input wavelengths). We used the quality-screened data available at the ORACLES ESPO Data Archive and presented in Pistone et al. (2019), but we further limit our analysis to SSA retrievals that had AOD (400 nm) > 0.4 similar to what AERONET uses in its quality-control criteria. Additionally, we only consider the SSA retrievals for which flight altitude was greater or equal to 1 km, which is about the typical boundary layer height over the ocean for the
ORACLES 2016 region. Since 4STAR retrieves the above-column SSA, our last screening criterion was used to emphasize on SSA of smoke layers above the boundary layer and exclude the influence of marine aerosols within the boundary layer.

We also use the atmospheric profiling provided by the NASA Langley Research Center High Spectral Resolution Lidar (HSRL-2, (Hair et al., 2008)) that was deployed on the ER-2 aircraft. Note that flight paths for the
P-3 and ER-2 only partly overlapped (Fig. 1a), for example on the September 24 case (Fig. 1b) that we analyze in greater detail to utilize this synergy between the two aircraft (Section 3.1). HSRL-2 measures aerosol backscatter and depolarization at 355, 532, and 1064 nm, and aerosol extinction at 355 and 532 nm. Out of the standard aerosol data products (Burton et al., 2012) that HSRL-2 provides, we mainly use the aerosol extinction and lidar ratio (extinction-to-backscatter ratio) profiles at 355 and 532 nm for this study. The lidar has a vertical resolution of 15 m, and it takes
measurements roughly every 0.5 seconds, but the data products were horizontally averaged or interpolated to the aircraft GPS times resulting in a horizontal resolution of about 1-2 km. Additionally, HSRL-2 has a higher signal-to-noise ratio than the space-based lidars, so it can robustly sample the complete aerosol vertical structure of the aerosol, especially above the marine boundary layer that is mostly capped by stratocumulus clouds in this region.

**Aerosol Composition**: The Single Particle Soot Photometer (SP2) was deployed on the P-3 as part of HiGEAR to
measure the mass of refractory black carbon (rBC) particles by passing a powerful laser beam and heating them to incandescence (Schwarz et al., 2006). The peak value of this incandescence signal has been shown to linearly correlate with the mass of the rBC particle (Stephens et al., 2003). The bulk submicron non-refractory aerosol composition, on the other hand, was provided by the time of flight (ToF) Aerodyne aerosol mass spectrometer (AMS) in the form of organic aerosol (OA), sulfate ($SO_4$), nitrate ($NO_3$), and ammonium ($NH_4$) mass concentrations (DeCarlo et al., 2008).
The AMS-measured mass concentrations are provided at standard temperature (273 K) and pressure (1000 hPa), but we convert them to ambient conditions using the ideal gas law and measured pressure and temperature information before comparing with the model equivalents.

**Aerosol Size Distribution:** Particle size distributions used in this study were measured with an ultra-high-sensitivity aerosol spectrometer (UHSAS, Droplet Measurement Technologies, Boulder CO, USA). UHSAS is an optical-
scattering, laser-based aerosol particle spectrometer that measures particles from 60–1000nm at 1s time resolution, thereby covering the entire accumulation mode. The UHSAS measured size distribution is reported as particle number concentrations (in cm⁻³) per size bins that are approximately logarithmically spaced.

**Carbon Monoxide (CO):** CO was measured with a gas-phase CO/CO2/H2O analyzer (ABB/Los Gatos Research $CO/CO_2/H_2O$ analyzer known as COMA). It uses off-axis integrated cavity output spectroscopy (ICOS) technology
to make stable cavity enhanced absorption measurements of CO, $CO_2$, and $H_2O$ in the infrared spectral region (Provencal et al., 2005). The measurements were reported as dry air volume mixing ratios in parts per billion (ppbv).

### 2.2    Ground-based Observations

The Aerosol Robotic Network (AERONET) measures the spectral aerosol optical thickness (AOT) through a ground-based network of sun/sky scanning photometers (Holben et al., 1998). For September 2016, Figure 1a depicts
the sites that were operational over southern African biomass burning source regions during ORACLES-I. AERONET provides spectral AOT to an accuracy of ±0.015 from direct sun measurements. In our analysis we use the Version 3,



Level 2 (cloud-screened, quality-assured) AERONET direct sun derived AOT product, specifically the AOT at 500 nm as well as the spectral SSA retrievals at 440, 675, 870, 1020 nm (Giles et al., 2019; Sinyuk et al., 2020).

## 2.3    Satellite Observations

### 2.3.1    OMI Absorbing Aerosol Index (AI)

We use space-based aerosol observations from the Ozone Monitoring Instrument (OMI; Levelt et al., (2006)) on board the NASA Aura spacecraft. OMI is a hyperspectral (270-500 nm), wide swath (~2600 km) imager that observes the back-scattered solar radiation with a nominal ground pixel size of 13x24 $km^2$ at nadir and 13x150 $km^2$ at the swath edges. Aerosol products are from the OMAERUV algorithm (version 1.8.9.1, after Torres et al., (2007)). Fundamental to the OMAERUV retrieval is the so-called aerosol index (AI). The AI is computed from the observed radiances at two channels where ozone absorption is weak (354 and 388 nm for OMAERUV), where the observed spectral contrast is compared to the (easily characterized) spectral contrast expected in a purely molecular atmosphere. Under cloud-free conditions the AI is sensitive to aerosol loading, height, and absorption, where increases in any of those quantities result in a higher positive magnitude value of the AI. OMI has near-daily global coverage and observes the sunlit portion of the Earth with a local equator crossing time of 13:30. Owing to its relatively large pixel size OMI data are frequently cloud-contaminated. We restrict our analysis to best available data, where the algorithm quality-assurance flag = 0. Additionally, OMI has suffered from a so-called "row-anomaly" since shortly after launch, in which a portion of its swath is physically obscured at the detector by an obstruction, resulting in reduced spatial coverage.

### 2.3.2    MODIS NNR (Neural Network Retrievals)

The Moderate resolution Imaging Spectroradiometer (MODIS) sensors have been flying on two spacecraft, Terra (10:30 local solar time equator crossing) since 2000 and Aqua since 2002. The MODIS collection 6.1 algorithms (namely, dark target land, ocean, and deep blue) retrieve aerosol properties over both ocean and land surface types using the observed spectral reflectance (Levy et al., 2013). However, instead of directly using the MODIS operational retrievals of aerosol optical depth (AOD) for model evaluation, we use a bias-corrected AOD dataset, called the MODIS NNR retrievals (Randles et al., 2017). NNR refers to a Neural Net Retrieval algorithm that computes AERONET-calibrated AOD from satellite-based radiances, in this case MODIS radiances. To derive 10-km resolution MODIS NNR AOD, over-ocean predictors include MODIS level-2 multichannel top-of-the-atmosphere (TOA) reflectances, glint, solar and sensor angles, cloud fraction (pixels are discarded when cloud fraction > 70%), and albedo derived using GEOS surface wind speeds. Over land, predictors are the same, except a climatological albedo is included for pixels with surface albedo < 0.15. The target of the NNR algorithm is the log-transformed AERONET AOD interpolated to 550 nm.

## 2.4    Model Description

Simulations are performed with a version of the NASA GEOS model (Molod et al. 2015), a global Earth system model used for near-real time weather and aerosol prediction, performing atmospheric analyses, and producing atmospheric and composition reanalyses, among other applications. The configuration of GEOS run in this study includes an atmospheric general circulation model run with the finite-volume dynamical core (FV3, after Putman and Lin, 2007) on a cubed-sphere horizontal grid. The model physics includes the Grell-Freitas deep convection (Freitas et al., 2018; Grell and Freitas, 2014) and Park-Bretherton, 2009 shallow convection, the Lock turbulence scheme (Lock et al., 2000), the catchment land surface model (Koster et al., 2000), RRTMG shortwave and longwave radiation (Iacono et al., 2008), and parameterized P-L chemistry (Nielsen et al., 2017). An aerosol-aware single moment cloud microphysics scheme is employed to provide dynamic cloud and ice effective radii (Bacmeister et al., 2006).

The aerosol scheme utilizes an updated version of the Goddard Chemistry, Aerosol, Radiation, and Transport (GOCART) module (Colarco et al., 2010; Chin et al., 2002). GOCART includes the sources, sinks, and chemistry of dust, sea salt, nitrate, sulfate, and carbonaceous aerosols. Dust and sea salt have dynamic (i.e., wind-driven) sources, and for each the aerosol particle size distribution is discretized into a series of five non-interacting size bins. Bulk sulfate mass is tracked, with primary emissions from anthropogenic sources and precursor emissions of dimethylsulfide (DMS), which has a wind-blown source function over the ocean, and sulfur dioxide ($SO_2$), which has emissions from anthropogenic, volcanic, and biomass burning sources. Chemical production of sulfate from precursors uses a simplified $OH-H_2O_2-NO_3$ scheme, with oxidants provided from the MERRA-2 GMI full-chemistry simulation (Strode et al., 2019). Nitrate is represented by three non-interacting size classes and has precursor emissions from anthropogenic, biomass burning, and ocean sources of ammonia. The fine mode nitrate includes heterogeneous formation from nitric acid on dust and seasalt surfaces (Bian et al., 2017). Carbonaceous aerosols are partitioned into



three compositional species—black carbon (BC), organic carbon (OC), and "brown" carbon (BR)—where each species is further divided into hydrophobic and hydrophilic modes. In the context of our simulations, "brown" carbon is organic carbon from biomass burning sources, separated from other anthropogenic and biogenic sources of organic aerosol to treat the optical properties distinctly (Colarco et al., 2017). A simplified secondary organic aerosol (SOA) mechanism is employed that scales volatile organic carbon (VOC) emissions in terms of carbon monoxide (CO) emissions from anthropogenic, biofuel, and biomass sources, following (Kim et al., 2015), with conversion of VOC to SOA a simple function of the MERRA-2 GMI-provided OH fields and the SOA going to hydrophilic modes of organic and brown carbon. Biogenic sources of SOA are from an online version of the MEGAN (Model of Emissions of Gases and Aerosols from Nature) mechanism running inside GEOS. Baseline optical properties are primarily based on the OPAC (Hess et al. 1998) database (see also Chin et al., 2002) with the optical properties for non-spherical dust particles based on Colarco et al., (2014) and the optical properties of brown carbon based on Colarco et al. (2017). The aerosol species are externally mixed for optics and chemistry purposes.

Biomass burning emissions in our simulations are based on the Quick Fire Emission Dataset (QFED; Darmenov and da Silva, 2015). QFED utilizes satellite fire radiative power observations from MODIS Level 2 fire products and scales them using biome-specific emission factors to aerosol and trace gas emission fluxes. In this study we particularly use QFED emission products for $SO_2$, CO, $NH_3$, BC, and organic aerosol (OA). Here and throughout the rest of the paper, model OA refers to organic aerosol contributions from all sources (biomass, biogenic, biofuel and anthropogenic) unless specifically mentioned.

Our baseline GEOS simulation is performed at a c360 (~25 km) horizontal resolution with 72 vertical hybrid sigma levels extending from the surface to ~80 km altitude. The model is "replayed" to the European Centre for Medium-Range Weather Forecasts (ECMWF) fifth generation atmospheric reanalysis (ERA5, Hersbach et al., 2020). "Replay" mode is like running the model in an atmospheric data assimilation mode, but instead of ingesting the meteorological observations directly we use the atmospheric state (surface pressure and vertical temperature, humidity, and horizontal wind components) from a prior analysis and compute the model incremental analysis update (IAU) from the forecast model background state versus that analysis, in this case versus ERA5. This approach allows us to perform a simulation constrained by realistic meteorology at a fraction of the cost of performing the full atmospheric data assimilation cycle. ERA5 was chosen to replay against versus the GEOS-native Modern-Era Retrospective analysis for Research and Applications, Version 2 (MERRA-2; Gelaro et al., 2017) because sensitivity simulations (Collow et al., 2023, *in preparation*) showed a more favorable representation of the smoke plume vertical distribution transported from southern Africa (see also Das et al., 2017) for a discussion of the challenges of models in representing the vertical profile of smoke in this region). Table 1 summarizes the suite of GEOS simulations performed in this study, described in subsequent sections of the text.

*Table 1. Summary of GEOS simulations performed in this study.*

| Simulation Name | Description |
|---|---|
| Baseline | Default version of the GEOS model |
| Smoke Age | Default version of the GEOS model run with biomass burning OA tagged by day of the week emitted (Section 2.4.1) |
| Smoke Composition | Default version of the GEOS model run with biomass burning OA tagged by type of vegetation burned (Section 2.4.2) |
| OA-loss | Hydrophilic OA from biomass burning is assigned a 6-day e-folding loss time; OA from biomass burning sources is enhanced 60%, BC from biomass burning sources is enhanced 15% |
| OA-loss+updated optics | As in OA-loss but with updated aerosol optical properties |

### 2.4.1 Estimation of Physical Smoke Age

Our "Smoke Age" simulation (Table 1) has the brown carbon tracer "tagged" in such a way as to determine the day of the week its emissions were injected on. Effectively, eight instances of the biomass burning organic aerosol are



tracked in this simulation (seven instances, one for emissions occurring each day of the week, and an eighth for the cumulative total emissions). The "Monday" tracer, for example, resets the tracer concentration globally to zero when the model clock ticks over to Monday at 0 UTC. The differences between the total and individual day-of-the-week tracers allows determination of the smoke age out to smoke emitted seven days previously. The simulation is otherwise
identical to our baseline simulation.

### 2.4.2 Estimating the Smoke Composition based on their Emission Source Vegetation Type

Our "Smoke Composition" simulation has the biomass burning OA tagged by the vegetation source burned based on the QFED inputs. QFED biomass burning emission sources are classified using the land cover or vegetation type information provided by the MODIS Land Cover Type Product (MCD12C1, IGBP classification scheme, Friedl
et al., (2010)) for the year 2016, shown in Figure 2. The MCD12C1 product supplies global maps of land cover at annual time steps at 0.05° spatial resolution in geographic latitude/longitude projection. Further details on the Collection 6 MODIS Land Cover products, including MCD12C1 are provided in their User Guide: https://lpdaac.usgs.gov/documents/101/MCD12_User_Guide_V6.pdf, last accessed: 27 April, 2023. We define six major vegetation or land cover (LC) types over central and southern Africa that could contribute to the smoke over
ORACLES region: (1) savannas (LC type 9), (2) woody savannas (LC type 8), (3) grassland (LC type 10), (4) tropical forest (LC type 2, 4, 5), (5) shrubland (LC type 6,7) and (6) croplands (LC type 12, 14) (Fig. 2). We then define seven instances of brown carbon tracers (six are "tagged" based on their emission source vegetation type, plus one for the total) within the model to be able to investigate the contributions of each of these vegetation types towards the smoke observed over the ORACLES region.

### 2.4.3 OMI AI Simulator

A challenge in comparing models to satellite observations involves the translation of the model's prognostic variables (in our case, aerosol mass by species) into the optical quantities retrieved from the satellite observations (i.e., AOD). In general, models and satellite algorithms apply different assumptions about the aerosol microphysical and optical properties, which adds uncertainty to the resulting comparisons. One approach to bring the models and
observations closer together is to the translate the model into something closer to what the satellite fundamentally observes, the spectral back-scattered radiation. To that end we have previously built a simulator of the OMI AI (Buchard et al., 2015; Colarco et al., 2017) that uses as input the GEOS-modeled aerosol mass distributions and meteorological fields and computes the OMI radiances (and hence the AI) under the OMI viewing conditions (viewing geometry, terrain height, and surface reflectance). The simulator thus places the burden of the comparison onto the
quality of the aerosol fields simulated by the GEOS model (i.e., the simulated aerosol spatial and compositional distributions, as well as the assumed optical properties). We constrain as best as possible the quality of the aerosol simulation with the ORACLES observations and check consistency of the resulting aerosol fields with the OMI observations, which have the potential to further constrain the model over broader temporal and spatial scales. A caveat on this approach is that OMI, with its large pixel sizes, is frequently partially cloud contaminated and we are
not attempting to simulate the cloud impact on radiances with the GEOS inputs. After the Colarco et al. (2017) study, the subsequent releases of the operational AI included additional corrections of the observed reflectances by estimating the possible cloud contamination and then subtracting the estimated cloud reflectance from the total observed radiance (Torres et al., 2018). An attempt to replicate the same computation using GEOS would need a model estimate of the cloud fraction in the pixel, which would add an additional uncertainty to the radiance computation. We therefore
perform our comparisons only for the highest quality OMI retrievals (formally, QA-flag = 0) to eliminate as much as possible cloud contamination issues. When using the high-quality pixels, the difference between the newest versions of the AI and the one used here are minimized to the point that in clear sky pixels both are almost the same (Hiren Jethva, personal communication). See Colarco et al. (2017) for further description of the AI simulator, including the specific description of the computation of the AI used here.

### 2.5 NOAA HYSPLIT Model with ERA-5 Meteorological Fields

We use the Linux-distribution of the NOAA (National Oceanic and Atmospheric Administration) Hybrid Single-Particle Lagrangian Integrated Trajectory (HYSPLIT, v5.2.1) model (Rolph et al., 2017; Stein et al., 2015) to understand the transport pathway and origin source locations of the smoke observed during ORACLES 2016. We use the ERA-5 meteorological data to drive the HYSPLIT trajectories so that the trajectories calculated are consistent with
our GEOS simulations. We also use the meteorological grid ensemble approach instead of single or center point trajectory approach within HYSPLIT to quantify the uncertainty and divergence associated with the trajectory calculations. In this method, trajectories are computed about a 3-dimensional cube centered about the starting point.



However, the initial positions are not offset, but just the meteorological data point associated with each trajectory, so that all trajectories start from the same point. This results in 27 members of the trajectory ensemble for all-possible offsets in X, Y, and Z directions in space.

## 3 Results and Discussions

In the following we analyze the ORACLES airborne data and our GEOS simulations first for the specific case of the aerosol plume flown by the P-3 and ER-2 on September 24, 2016. Refinements to our simulations are then described. We conclude by summarizing our simulation of the entire month of September 2016 in context of ground-based and space-based remote sensing observations and estimate the total aerosol radiative forcing during the month.

### 3.1 24 September 2016 Case

We focus here on the September 24, 2016, smoke plume observed by both the P-3 and ER-2 aircraft (Figure 1b). We choose this day for two main reasons: (1) the presence of multi-layered smoke around 12°S and 11°E that showed a gradual but substantial change in in-situ measured SSA vertically (from 0.83-0.91 over 1-5.5 km respectively) that GEOS was unable to capture in our baseline simulation, and (2) to utilize the synergy of multiple instruments onboard both the P-3 and ER-2 aircraft as they sampled the same locations for a significant part of their flight. We suggest the vertical variation of SSA is due to differences in the smoke age sampled on this day, as has also been discussed in detail by Dobracki et al. (2023) from a primarily observational perspective.

On September 24 the P-3 flew north-south along 11°E longitude to-and-from Walvis Bay, Namibia (Fig. 1b). The P-3 vertical flight trajectory is shown in Fig. 3 with the in-situ (PSAP and nephelometer) measured dry extinction superimposed on the baseline GEOS-simulated extinction profile along the flight track. The blue stars on Fig. 3 indicate the location of 4STAR sky-scans for which quality screened column-integrated SSA were retrieved (Pistone et al., 2019). On the same day the ER-2 aircraft carrying the HSRL-2 lidar spatially overlapped with part of P-3 flight path (Fig. 1b). The retrieved and GEOS-simulated vertical aerosol extinction profile (GEOS sampled along the ER-2 track) are shown in Fig. 4. A similar aerosol plume structure is apparent in GEOS comparisons to these two sets of observations (Figs. 3 and 4).

At about 13 UTC, the P-3 sampled a multi-layered smoke plume around 12.3°S and 11°E while descending from about 6 km to 1 km. The ER-2 flew over the same location earlier in the day (~ 9.5 UTC). This common spatial region is shown by the black rectangular box in Figs. 3 and 4 and is indicated by the green star in Fig. 1b. We compared the observed aerosol extinctions for this profile based on in-situ instruments (Fig. 5a) and HSRL-2 lidar (Fig. 5b) with our GEOS baseline simulation. Since the in-situ observations are at dry RH (<=40%) and the lidar observations are at ambient conditions, the differences between the observed extinction magnitudes can be explained in part by the high RH values (~80%, Fig. 5c) that caused enhanced scattering for the top aerosol layer centered around 4.5 km. GEOS simulated dry extinction is underestimated compared to in-situ data, but has a better match with HSRL-2 retrieval of ambient extinction, at least for the upper level smoke layer. The model simulated ambient extinction is underestimated for the lower aerosol layers despite having a very good match of simulated RH with the observations overall (Fig. 5c).

We compare the observed and simulated SSA on this profile in Fig. 5d. The in-situ measurements of dry SSA at 550 nm showed a gradual decrease in values from about 0.92 to 0.84 between the altitudes of 5.5 km to 1 km respectively. The model simulations of SSA, on the other hand, showed some decrease from the aerosol layer centered around 4.5 km to the lower altitude layers but stayed fixed at about 0.89 from 4 km to 1 km altitude. Clearly the baseline GEOS model does not simulate the observed vertical variability in SSA.

#### 3.1.1 Composition, SSA and Relation with Simulated Smoke Age

To understand the vertical variations in SSA, we first analyze the smoke composition of this September 24 profile by comparing the model simulated aerosol mass concentrations to the corresponding measurements from the AMS and SP2 in-situ instruments (Fig. 6). Organic aerosols (OA, from all sources) contribute about 65-70% of the total aerosol mass, followed by nitrates, sulfate and BC, respectively in both model and the observations. Model simulated aerosol composition shows a very good match with in-situ observations, especially for the top plume centered around 4.5 km. OA is overestimated by the model for altitudes below 2.5 km, while nitrates are underestimated for the same altitudes. Sulfate and BC are slightly underestimated in the model overall, but the multi-plume structure of the observed smoke profile is very well captured by the model.

SSA strongly depends on the relative amounts of scattering to absorbing aerosols. Here, BC is the primary absorbing component at 550 nm, while OA, nitrates and sulfates are mostly scattering. Given that the SSA is changing





with altitude and multiple plumes are observed within the vertical profile, we investigate two things: (1) the smoke age in the vertical, and (2) how the relative aerosol composition varies with altitude and smoke age. Figure 7a shows the model simulated, extinction-weighted mean smoke age for our September 24 profile using our "Smoke Age" tagged tracer run (Table 1, Section 2.4.1). We observe an almost monotonic increase in smoke age with decreasing altitude below 5 km. The simulated smoke age profile shows a younger smoke layer (4-5 days old) centered at 4.5 km followed by older smoke at the lower levels, with the oldest smoke layer (7-8 days old) between 1-2 km. Figure 7a also shows the smoke age derived using WRF-AAM (Weather Research and Aerosol Aware Microphysics) Model (Saide et al., 2016) that was used for forecasting and flight planning during the ORACLES campaign (Redemann et al., 2021), demonstrating that the two models are in close agreement. We also show the smoke age distribution from our GEOS "Smoke Age" run at three different altitudes of the profile to clarify the interpretation of the mean smoke age number and to note some of the limitations of model-calculated smoke age (Fig. 7b). For example, for the youngest smoke plume centered at 4.5 km, even though the mean smoke age is calculated as 4.1 days, about 40% of the smoke is 1-2 days old and about 20% of the smoke is older than 7 days, suggesting quite a wide distribution. At 3 km altitude the lower mode of the smoke age distribution peaks at 2-3 days old, and at 2 km at 3-5 days old. We should also be mindful that different age distributions could lead to a same mean age value. Finally, for GEOS, we are restricted by the way we track the smoke age that we can only resolve smoke age up to past 7 days. For smoke older than 7 days, a single weight is provided, effectively causing the weighted mean smoke age to be slightly younger than it possibly is. Nevertheless, the simulation shows presence of generally younger smoke at the higher altitudes and older smoke at the lower altitudes.

Next, we investigate how the relative composition of aerosols change with smoke age, and if we can further use that relation to explain the vertical variations of SSA as well. Figure 8a shows the simulated and observed change in SSA with the GEOS-simulated mean smoke age for the profile shown in Fig. 5d, and this variation is almost identical to SSA variation in the vertical since smoke age and altitude are strongly but negatively correlated. The initial drop in SSA between 4-5.5 days estimated smoke age is correlated with a simultaneous decrease in OA and nitrates with respect to BC (Figs. 8b&c). After 5.5 days age the nitrates are essentially constant in time, and so further decrease in SSA with smoke age is more clearly correlated with a continued decrease in OA:BC ratio (Fig. 8b). The model captures the variation in nitrate:BC and SO$_4$:BC ratio well for most parts of the profile (Fig. 8c&d). The major discrepancy between the model and observations lies in the inability of the model to simulate the observed OA:BC variation, which changes in the observations from about 15 (for 4-day old smoke) to about 5 (for 7-day old smoke), whereas the model is essentially constant at a OA:BC ratio of 15. The model's failure to capture this behavior is strongly correlated with its failure to capture the observed variability in SSA.

In the following we consider two possible explanations for the observed variation of SSA and OA:BC ratio with age toward overcoming the shortcomings of the model simulation.

### 3.1.2 Are Smoke Layers Originating from Different Emissions Sources or Burning Conditions?

The first hypothesis we examine is whether smoke of different ages is originating from different source regions with perhaps different characteristics in the composition of emitted species or different proportions of flaming to smoldering phase of the combustion products. We use three approaches to examine this hypothesis: (1) we generate ensembles of back-trajectories to track the origin locations of smoke layers at different vertical levels, (2) we utilize the vegetation type "Smoke Composition" tracer run of our GEOS model (Table 1, Section 2.4.2) to quantify the contribution of each vegetation type towards OA amounts (or extinction), and (3) we examine the observed black carbon to carbon monoxide ratios as an indicator of whether the smoke arises from flaming versus smoldering combustion. Our analysis is for the same September 24 aerosol profile discussed previously.

Figure 9a shows the emission locations and cumulative amounts of OA emitted over the seven days prior to September 24. Figures 9b&c show the HYSPLIT-model generated ensemble back trajectories (multi-colored lines) originating from two different vertical levels (4.5 km versus 2 km) representing the location of the two distinct smoke plumes evident in the profile plots (e.g., Fig. 6a). For the higher-altitude smoke layer (centered around 4.5 km), the back trajectories (Fig. 9b) show that there is combined contribution from smoke originating from the emission locations (Fig. 9a) that are north, south, and far east of the smoke profile (at 12°S, 11°E). The trajectories further suggest that the contribution of smoke from near coast burning sources located north and south of the profile are possibly causing the higher-level smoke to be younger (about 4 days old, Fig. 7). The lower-altitude initialized back trajectories (originating at 2 km, Fig. 9c) on the other hand, suggest that the smoke got transported almost directly east-to-west across the continent with a significant number of trajectories ending over the far-east burning locations (Fig. 9a), thereby making the smoke at these levels older (about 6-7 days old, Fig. 7).





The trajectory information alone does not tell us that the smoke origins (or OA:BC ratio at emissions, as a proxy for vegetation type) for the distinct smoke layers are similar. Therefore, we use the "Smoke Composition" simulation to quantify and compare the contribution of the six major vegetation types of this region towards the smoke amounts (or OA extinction) for the same higher and lower-level smoke layers (Fig. 10a). Our analysis suggests that the major
emission source for both the smoke layers include savannas, woody-savannas, and grasslands. The contributions from the other three vegetation types (tropical forests, shrublands and croplands) amount to only about 20% of the total emissions. More importantly, the contributions from all vegetation types, except grasslands and woody-savannas, are comparable for both the higher and lower-level smoke plumes. The higher grassland contributions (relative to woody-savannas) for the 2 km level are possibly due to smoke originating from the grassland-dominated regions south of
15°S and east of 20°E based on the 2 km back trajectories (Fig. 9c) and the land cover map (Fig. 2). However, both Akagi et al. (2011) and Andreae, (2019), which are the key databases that most global models use for their assumptions of emission factors (or emission ratios) of different biomass burning species, do not even differentiate between grasslands and savannas as fuel types, assigning them both the same emission factors. Based on these databases, and in terms of the organic carbon (OC)-to-BC ratios, the most different fuel types for this region compared to
grasslands/savannas are crop or agricultural residue and tropical forest (Table 2), but both of these fuel type categories appear to have minor and almost equal contributions at the two vertical levels considered here (Fig. 10a).

*Table 2. Emission ratios for different fuel types relevant to southern African biomass burning region based on two primary databases that most global models use.*

| Biomass Burning Fuel Types | Mean OC:BC Emission ratio | OA:BC (=1.8*OC:BC) | References |
|---|---|---|---|
| Savanna and Grasslands | 7.08 | 12.74 | Akagi et al. (2011) |
| | 5.67 | 10.2 | Andreae (2019) |
| Tropical Forest | 9.06 | 16.3 | Akagi et al. (2011) |
| | 8.63 | 15.5 | Andreae (2019) |
| Agricultural or Crop residue | 3.07 | 5.52 | Akagi et al. (2011) |
| | 11.70 | 21.0 | Andreae (2019) |

Finally, differences in OA:BC ratio could also be due to different combustion characteristics of the source fires
(i.e., flaming versus smoldering), wherein flaming fires are known to generate a lower concentration of organic carbon particles but a higher concentration of BC than smoldering fires (Christian et al., 2003; Yokelson et al., 2009). We note here that we currently do not make the distinction in emission ratios based to fire characteristics within QFED or GEOS. Vakkari et al. (2018) suggests that excess BC to excess carbon monoxide ratio ($\Delta BC/\Delta CO$ or BC:CO here onwards) can be used as a reliable marker to assess the combustion characteristics, especially for diluted and aged
plumes. An increasing $\Delta BC/\Delta CO$ value is indicative of increasing flaming fraction during the burning (Yokelson et al., 2009). Precisely, Vakkari et al. (2018) distributed the BC:CO ratios into bins of 0.005 to represent fires of similar characteristics. Further, based on their measurements over southern African savannah and grassland region, they found that BC:CO < 0.005 represents predominantly smoldering conditions, while BC:CO > 0.010 represents predominantly flaming conditions. We consider background CO as 100 ppbv based on the outside plume (> 6 km) values of CO in
Fig. 10b for the September 24 case. Figure 10c shows that the excess BC:CO ratio has very little variation in the vertical and the values are within the range of 0.005-0.010, thereby suggesting approximately equal mix of smoldering and flaming combustion conditions. Therefore, it is unlikely that the differences in smoke composition (or OA:BC ratio) at different vertical levels occurred due to differences in burning source vegetation type or combustion conditions.

### 3.1.3 Implementation of Additional OA Loss Rate

The second hypothesis we consider is that there is a chemical or microphysical loss of OA during transport that is related in some way to the smoke age. OA from biomass burning sources is composed of primary organic aerosol (POA) directly emitted from the burning biomass and SOA formed via oxidative processing of organic gases. As the



biomass burning plume ages in the atmosphere, the oxygenation of OA (reflected in O/C ratio) is reported to increase
significantly in almost all studies, suggesting strong chemical transformation of OA with aging and formation of more
oxidized OA that are compositionally different than POA (Capes et al., 2008; Jolleys et al., 2012; Cubison et al., 2011;
Forrister et al., 2015; Zhou et al., 2017). OA mass is increased by SOA formation but can also be balanced out by
dilution and subsequent evaporation to the gas phase of semivolatile components, resulting in loss of POA mass
(Cubison et al., 2011; May et al., 2015; Zhou et al., 2017). In addition, the volatilities of organic species are affected
by atmospheric oxidation reactions. In the gas phase, two main processes affect the volatilities of organics:
fragmentation and functionalization. Fragmentation leads to a loss of carbon from the particle (assuming at least one
of the fragments is volatile), whereas functionalization leads to an increase in particulate oxygen. In the condensed
phase, additional bimolecular processes, such as accretion/oligomerization reactions can also affect volatility (Kroll
et al., 2009). Changes of OA composition with aging inevitably lead to changes to OA optical and hygroscopic
properties and thus have climate implications. The reasons for the variability across studies may be related to variations
in fuels burned and combustion conditions, variation in key co-emitted species such as NOx, and differences in
environmental conditions such as dilution rate and temperature and humidity (Shrivastava et al., 2017).

Within GEOS, the OA:BC ratio is prescribed at emissions based on the burning fuel type and related emission
factors. Further, OA is emitted as 50% hydrophobic and hydrophilic, respectively, while BC is emitted as 80%
hydrophobic and 20% hydrophilic. Once emitted, both OA and BC undergo hydrophobic-to-hydrophilic conversion
with a conversion rate or e-folding time of 1-2 days (Colarco et al., 2010). This hydrophobic-to-hydrophilic conversion
can change the OA:BC ratio somewhat from what it is at emissions due to wet scavenging, but for further aged smoke
(> 2 days old), we do not account for any additional OA or SOA loss due to any microphysical or chemical processes.
Indeed, our simulated OA:BC ratio remains mostly constant over time, as in Fig. 8b.

With the present model limitations in mind, we introduce an ad hoc loss process for the hydrophilic OA in our
model. We additionally increase our overall emissions of biomass burning produced OA to compensate for this added
loss channel to approximately preserve the overall regional extinction and AOD. Sensitivity studies suggest a six-day
e-folding time applied to the biomass burning hydrophilic OA, with a corresponding increase of about 60% in
emissions of OA and about 15% in the BC emissions from biomass burning greatly improves our simulated OA profile
(Fig. 11a). When these factors are applied to our simulations ("OA-loss" simulation, Table 1), we can simulate well
the observed SSA variation with altitude (Fig. 11b). Simultaneously, these changes resulted in overall less scattering
(or more absorbing) aerosol mixtures and brought the modeled SSA curve closer to 4STAR observations (Fig. 11c).

### 3.2   Updated Optics: Adjusting OA Size Distribution, Hygroscopicity, and Absorption

The previous section focused on the simulation of the aerosol lifecycle and mass distributions with emphasis
on the profile flown on September 24, 2016. Here we broaden our use of the ORACLES dataset to better constrain
the simulation of optical properties. As briefly described in Section 2.4, the model-derived aerosol optical properties
(e.g., extinction coefficients, SSA and phase function) for each aerosol component depends on the assumptions of
their microphysical (size distribution, refractive index etc.) and hygroscopic (or water uptake) properties. Therefore,
we use the ORACLES dataset to first evaluate and then adjust three key model assumptions in this regard to improve
the model agreement with the observations. Since OA contributes to about 60-70% of the smoke plume mass observed
during ORACLES, only the particle properties of biomass burning OA/brown carbon component in the model were
adjusted.

The first adjustment was made with respect to the particle size distribution. Within GOCART, a single-mode
lognormal size distribution for the OA component is assumed, defined by a number mode radius ($r_n$) and geometric
standard deviation ($\sigma$). The baseline model assumption for OA number size distribution (Fig. 12, red dashed line)
shows a $r_n$=0.021 nm and $\sigma$=2.2 based on Chin et al. (2002). The comparison with UHSAS-measured particle size
distribution for the available flight days shows that the observed particle size distribution (for accumulation mode)
has a larger mode radius and narrower distribution than the model baseline assumption and does not show much
variation amongst different flight days. Nonetheless, we limit the observations to the altitudes between 1-5 km and
OA concentration greater than 4 mg/m$^3$ to ensure the presence of smoke particles. Therefore, for the updated optics
case, we fit a lognormal distribution to the UHSAS-measured size distributions with a $r_n$ =0.09 nm and $\sigma$=1.5 (Fig.
12).

The second change is related to the model assumption of particle hygroscopicity. We utilized the dry and
humidified nephelometer-measured scattering coefficients during ORACLES-I to obtain the scattering enhancement
factor, $f$(RH). The $f$(RH) is defined as the particle light scattering coefficient at elevated RH, divided by its dry value.



We consider 10% and 80% as the low and high RH, respectively, and hence consider only those measurements that were obtained closest to these RH values before comparing with the model-derived *f*(RH) values at the same flight location (Fig. 13a). Additionally, to ensure the presence of smoke plumes, we only consider measurements that were made between 1.5 to 6 km altitude levels. Figure 13a showed that GEOS baseline case significantly overestimates the observed *f*(RH) by almost a factor of 2. This model overestimation in *f*(RH) has also been previously reported in multi-model evaluation studies of Doherty et al. (2022) using the ORACLES dataset itself, and in Burgos et al. (2020) using a global dataset of surface-based in situ measurements. Note that the modeled *f*(RH) presented here has contributions from a combination of smoke aerosol species. For GOCART, except for dust, all aerosol components are considered to have different degrees of hygroscopic growth rate with ambient moisture. We show the hygroscopic growth assumption for three major smoke aerosol components (organics, black carbon, and sulfate) in Fig. 13b based on the default optics table used within GOCART. Sulfate hygroscopicity increases most rapidly with increasing RH, followed by organics and least rapidly for black carbon (or BC). Therefore, for the "OA-loss" model case (Section 3.1.3), where relative contribution of BC is enhanced for the aged plumes compared to OA, we see a slight reduction in modeled *f*(RH) compared to observations (Fig. 13a). An additional, but minor reduction in modeled *f*(RH) is observed when the above-mentioned particle size distribution adjustments are made for the organic aerosol component in the model. Finally, we consider that the hygroscopic growth of OA tracks with BC, and here we see the closest match to observations, and so this comprises our second adjustment to model assumption of OA properties.

The third and final adjustment to the model optical properties is related to aerosol absorption, which is mainly characterized by the imaginary part of the complex refractive index (k) of the aerosol component. This change was motivated by our finding that our model-derived lidar ratios of the smoke plumes were much higher than the observed values based on HSRL-2 measurements from the ER-2, especially in the blue (or 355 nm) channel (70 sr versus 85 sr) for the model baseline case (Fig. 14a). A previous study by (Veselovskii et al., 2020) that used the Raman Lidar observations over Senegal in West Africa for their analysis, also found that the GEOS modeled lidar ratios were consistently higher compared to their lidar observations for a range of relative humidity, when the model k in the UV were set to its baseline value (of $k_{350}$ ~0.05). The aerosol lidar ratio is defined as the ratio of its extinction-to-backscatter coefficient, or more precisely, as the ratio of the volume-extinction cross section (scattering in all directions plus absorption) to the 180° volume-backscatter cross section of the aerosol particles. The aerosol backscattering coefficient is sensitive to the absorption, but the extinction coefficient is only weakly dependent on it because an increase in absorption is accompanied by a decrease in scattering. Thus, we expect that decreased absorption of OA in the UV should lead to a decrease in the lidar ratio. Figure 14 demonstrates the sensitivity of the lidar ratios to the changes in the imaginary component of the refractive index. The baseline optics table assumes $k_{350}$ = ~0.05, and so clearly reducing $k_{350}$ to 0.01 or 0.02 provides a better match of the model lidar ratio with the HSRL-2 observations (Fig. 14a). Note that we preserve the spectral contrast for all wavelengths < 550 nm, so our adjustment of $k_{350}$ also implies a change at the longer wavelengths, hence we simulate a different lidar ratio at 532 nm (Fig. 14b). For our final set of optics, we chose $k_{350}$ = 0.02 to have a reasonable simulation of the OMI Aerosol Index (AI) values that we discuss later in Section 3.4.2. It is also worth mentioning that lidar ratio also depends on the size distribution and shape of the particles (Meng et al., 2010). We have constrained the particle size distribution using ORACLES observations, as discussed earlier, but GOCART still assumes its smoke components (OA, BC, NI and SU) to be spherical particles only.

A new set of optics table was generated following the above three changes in assumptions of OA microphysical and water uptake properties that we call as "updated optics" throughout our figures and text. Now, we can revisit Fig. 11b&c to understand the impact of these optics update on the model simulated SSA comparisons with the in-situ and 4STAR observations. For 550 nm, there is only a slight decrease in model SSA with "updated optics" compared to the "OA-loss with default optics" case and is yet very close to the observations (Fig. 11b). Figure 11c is better at showing the impact of these changes on the broader spectral curve. The "updated optics" case is less absorbing in the shortest 400 nm channel, but otherwise more absorbing than both the baseline and "OA-loss with default optics" case. The former is expected due to the reduced OA absorption in near-UV and latter is possibly due to the reduced hygroscopic growth assumptions that is leading to the formation of effectively smaller particles at enhanced RH within the smoke plumes. However, overall, the "OA-loss+updated optics" case shows a better match with the 4STAR observations compared to the other two cases. Note that the results in Fig. 11c are the average of three 4STAR retrievals on Sep 24, 2016 (Fig. 3), but similar SSA magnitudes and spectral curve shape, both for 4STAR and the different model simulations hold when we average across all days of the ORACLES-I campaign (not shown here).



### 3.3 AOD and SSA Evaluation at Near-Source AERONET sites

We evaluated the model performance of AOD and SSA at all the AERONET sites depicted in Fig. 1a. For brevity however, we present here the comparative results from two key near-source sites, Mongu_Inn and Lubango, that had data availability for most days of the September 2016 and are most contrasting in terms of the model performance. The model shows a very good agreement with AERONET observations over Lubango in terms of both AOD and SSA, and the final OA-loss+updated optics case does not deviate much from the model baseline case (Fig. 15a, c). At Mongu_Inn on the other hand, the model appears to have a systematic low bias compared to the AERONET observed AOD but nonetheless can capture its daily variability (Fig. 15b). For SSA at Mongu (Fig. 15d), the model largely overestimates the absorption at longer wavelengths, and this together with AOD underestimation suggests that model is likely missing a contribution from coarse-mode particles over this site. However, the model "OA-loss+updated optics" case here corrected for the excessive absorption at the shortest AERONET channel compared to the baseline simulation.

### 3.4 Satellite Perspective: Impact of updated optics on AOD and AI

#### 3.4.1 Comparisons with MODIS NNR AOD Retrievals

Satellite-based observations are used in this section to assess the model performance over complete SE Atlantic as well as the source region over southern and central Africa. Figure 16a shows the September 2016 monthly mean MODIS NNR AOD. We sample the model AOD based on the MODIS swath, overpass time, and data availability, and the resulting monthly mean AOD from the baseline, OA-loss, and OA-loss+updated optics simulations are shown in Fig. 16b, d, and f, respectively. The last column (Fig. 16ceg) shows the differences between the Model and MODIS NNR AOD for the three simulations. Overall, there is a good match between the model and NNR retrievals over the ocean and the smoke outflow region compared to the source region over the continent. There is a significant underestimation in terms of model AOD over the south-east parts of the continent that persists in all the three model simulations. This is a shortcoming of GEOS model that has persisted in previous model versions as well, possibly due to missing biomass burning or coarse-mode aerosol sources (Das et al. 2017). Otherwise, there is a high bias in the model AOD close to the west coast along 10ºS in the baseline simulation (Fig. 16c). In the intermediate model simulation, called "OA-loss" (Fig. 16e), this high bias in model AOD along 10ºS slightly decreases, especially over the ocean, but persists over the continent. This is expected due to the accounting of the additional loss of OA with increasing smoke age as the plumes move away for the continental biomass burning sources. The final simulation with "OA-loss+updated optics" (Fig. 16f, g), however, appears to perform the best amongst the three model simulations by overall mitigating the high model AOD bias along 10ºS compared to the observations. Note that we chose to show only the Aqua MODIS results here since Aqua has a closer overpass time with OMI (Aura) than Terra, but the comparisons of model AOD simulations with respect to Terra MODIS retrievals (not shown here) also showed similar results.

#### 3.4.2 Comparisons with OMI AI Observations

In Fig. 17 we show the GEOS-simulated OMI aerosol index (AI) for September 2016 in comparison to the OMI retrievals. The GEOS modeled aerosol profiles at the OMI footprints are sampled and, along with the model optical property assumptions, provide one set of inputs to the AI calculator. The other inputs are the OMI observation geometry and retrieved surface reflectance. To minimize the impact of clouds on the comparison we retain only OMI pixels with QA=0 (low probability of cloud contamination). The OMI retrieved AI is shown in Figure 17a. The simulated radiances from which the model AI is calculated only include terms for the aerosol, molecular background, and surface. Two cases are shown in the comparison: the GEOS baseline run (Fig. 17b) and the final run that includes OA loss with smoke age, increased biomass burning emissions, and ORACLES-informed optical properties for OA (OA-loss+updated optics, Fig. 17d). The baseline run with its more highly absorbing OA assumptions and high bias in aerosol loading especially near the coast of southwest Africa (10ºS, Fig. 16c) results in the model overestimating the AI relative to OMI, with this high bias especially apparent over both land and ocean where the AOD was overestimated in the model (Figure 17c). By contrast, the GEOS model run with the ORACLES-informed adjustment to the optical property assumptions, along with the reduced overall burden of OA, results in a much more favorable comparison of the simulated to retrieved AI (Figure 17e). This assessment of the model performance with the independent OMI dataset shows the overall consistency between the in situ derived optical properties from the ORACLES measurements and what can be retrieved from space-based remote sensing, with the GEOS model as the interpolator between the two datasets.





### 3.5 Radiative Implications

Finally, we consider the impact of the updated aerosol simulation on the direct radiative forcing of the aerosols, which along with any possible aerosol-cloud interactions drive the aerosol impacts on climate. Figure 18a shows the September 2016 monthly mean 550 nm AOD difference between the GEOS baseline and updated (OA-loss+updated optics) model runs. The difference is consistent with the results in Fig. 16 and shows that the enhanced biomass burning emissions in the updated run result in a slightly higher AOD over southeastern Africa, while the inclusion of the OA loss process in the updated run leads to somewhat less aerosol downwind and an overall lower smoke AOD over the southeast Atlantic compared to the baseline.

Figure 18c shows the top-of-atmosphere (TOA) all-sky shortwave (SW) radiative forcing due to aerosol difference between our two runs, computed in the model by two calls to the radiative transfer code, one with and the other without the aerosols. The aerosol forcing is defined as the difference between the net TOA radiation with aerosols minus the net TOA radiation without aerosols (i.e., $SW_{net\_aer} - SW_{net\_noaer}$) and represent an input (positive) or reduction (negative) of energy. There is an overall ~1 W m$^{-2}$ difference in the SW forcing over the southeast Atlantic on the north edge of the smoke plume. This difference is on the margins of the smoke plume to the north of the main cloud features (Figure 18b, shown for the Baseline run, similar results for the updated run) and has a negative magnitude as defined because in the baseline simulation there is more (relatively cooling) aerosol over the darker ocean surface and so more reflected radiation to space. The broad positive SW forcing difference over the continent centered near the border of the Democratic Republic of Congo and Zambia reflects the relatively greater backscattered solar radiation over the dark continental surface corresponding to the higher AOD in that region in the updated model run.

Finally, Figure 18d shows the SW atmospheric heating due to aerosols in the two model runs. This is the difference in the net forcing at TOA and at the surface. That enhanced loading (Figure 18a) and the relatively more absorbing optics of the baseline suggest a greater warming of the atmosphere versus the updated run, which would tend to stabilize the atmosphere. By contrast, our updated simulation has somewhat less atmospheric warming over the regional cloud decks.

### 4 Conclusions and Future Directions

In this study, we utilized the detailed ORACLES-I aircraft observations to evaluate the current state of biomass burning aerosol properties and transport in the NASA GEOS model following recent developments in its representation of carbonaceous aerosols. An example case study of September 24 was analyzed in detail, during which in-situ and remote sensing observations showed a presence of multi-layered smoke plume structure with significantly varying SSA in the vertical. Our baseline simulations were not able to represent the observed OA:BC ratio nor the vertically varying single scattering albedo. Analyzing the simulated smoke age suggests that the higher altitude, less absorbing smoke in the plume was younger (~4 days), while the lower altitude and more observing smoke plume was older (~7 days). We hypothesize a loss process—chemical or microphysical—explains the change in aerosol absorption as the smoke plume ages, and we apply a simple 6-day e-folding loss rate to the hydrophilic biomass burning OA to mimic this process. Adding this loss channel requires some adjustment to the assumed aerosol emissions to approximately conserve the regional aerosol loading and further improve the simulated OA:BC ratio, and accordingly we have increased our biomass burning emissions of OA by 60% and biomass burning BC emissions by 15%. Next, we use the aircraft-based observations to better constrain the simulation of aerosol optical properties. For the biomass burning OA aerosol component within GEOS, we adjust the lognormal size distribution to have a modal radius, $r_n$ =0.09 nm and $\sigma$=1.5 based on the ORACLES-I HiGEAR (or UHSAS) observations. Secondly, the OA hygroscopic growth is assumed to increase less rapidly with increasing RH compared to default assumptions and mimic the hygroscopic growth rate of model BC, which provides a simulated $f$(RH) similar to the observed. Finally, we also adjusted the OA complex refractive index ($k_{350}$=0.02) to reduce the OA absorption in the near-UV to match better with the HSRL-2 retrieved lidar ratio and OMI retrieved AI. Our final GEOS model simulation with additional OA loss and updated optics showed a better performance in simulating AOD and SSA compared to independent AERONET observations and MODIS NNR AOD retrievals for the entire month of September 2016. In terms of radiative implications of our model adjustments, the final GEOS simulation suggests a decreased atmospheric warming of ~2 W m$^{-2}$ over the South-east Atlantic region and above the stratocumulus cloud decks compared to the model baseline simulations.

In terms of future directions, the simplistic approach presented in this study to address the microphysical or chemical loss of OA with increasing smoke age lacks a specific physical basis. In subsequent modeling studies, the mechanisms causing this OA loss can be further explored and if the OA loss is indeed associated with continued



oxidation of OA during the aging of smoke, the loss can be parameterized as a function of oxidant fields in the model rather than assuming a single e-folding time to mimic the aerosol loss. Moreover, observations over other major biomass burning regions of the globe need to be analyzed to find if a similar variation in SSA is observed for aging plumes in those regions and a global parameterization for OA loss will need to be explored. Finally, we will also need to utilize the observations from subsequent years of ORACLES (i.e., 2017 and 2018) to confirm whether the suggested set of OA microphysical and hygroscopic assumptions in the model for the 2016 case are also able to capture the seasonal variability of smoke emissions in this region.

*Code availability.* The GEOS Earth System Model source code and the instructions for model build are available at https://github.com/GEOS-ESM/GEOSgcm/ (Last accessed: 16 May 2023).

*Data availability.* The GEOS model outputs and MODIS NNR and OMI AI retrievals needed to reproduce the results described in this paper are publicly available for download at data.nasa.gov repository (https://doi.org/10.25966/wc4c-ke45, Das, 2023), see complete dataset citation under References section. The ORACLES data was obtained from their ESPO Data Archive: https://espo.nasa.gov/oracles/archive/browse/oracles/id8, last accessed: 22 March 2022. The AERONET AOD retrievals (version 3) and SSA inversion product were downloaded from https://aeronet.gsfc.nasa.gov/, last accessed: 24 April 2023.

*Author contributions.* SD and PRC conceptualized the modeling and analysis approach, performed the simulations, and wrote the manuscript. HB, SD, and PRC performed the model-data comparative analysis. PRC and SG performed the simulations of OMI AI and helped with the interpretations of the remote-sensing observational data. All authors contributed to the editing of the manuscript.

*Competing interests.* The authors declare that they have no conflict of interest.

*Acknowledgements.* We would like to acknowledge the NASA Earth Science Division and GEOS model developmental efforts at GMAO for their support. This work was supported by NASA's Aura Science Team award 19-AURAST-0014. The computing resources supporting this work were provided by the NASA High-End Computing (HEC) Program through the NASA Center for Climate Simulation (NCCS) at the Goddard Space Flight Center. The authors also gratefully acknowledge the NOAA Air Resources Laboratory (ARL) for the provision of the HYSPLIT transport and dispersion model used in this publication.

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





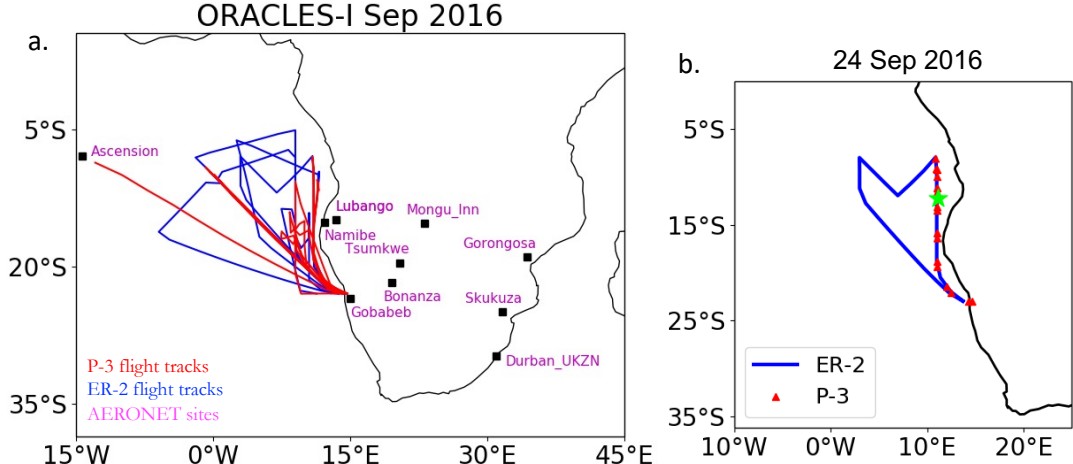

**Figure 1. (a) Location of ORACLES-I P-3 and ER-2 flights off southwestern Africa during September 2016 and ground-based AERONET sites over the continent. (b) P-3 and ER-2 flight tracks for September 24, 2016, with the green star**
**highlighting the location of the vertical profile illustrated in later plots.**



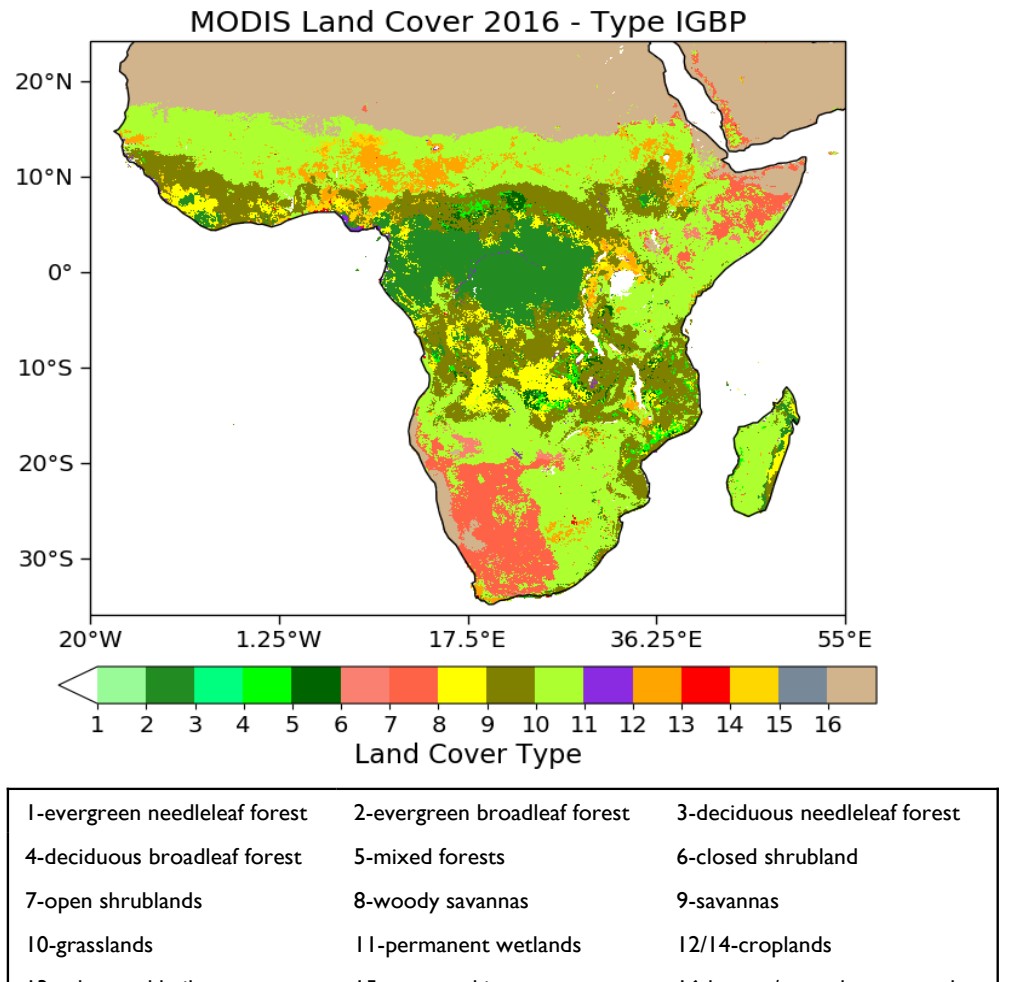

**Figure 2. Land cover or vegetation types for 2016 from the MODIS Land Cover Type Product (MCD12C1) using the IGBP classification scheme.**




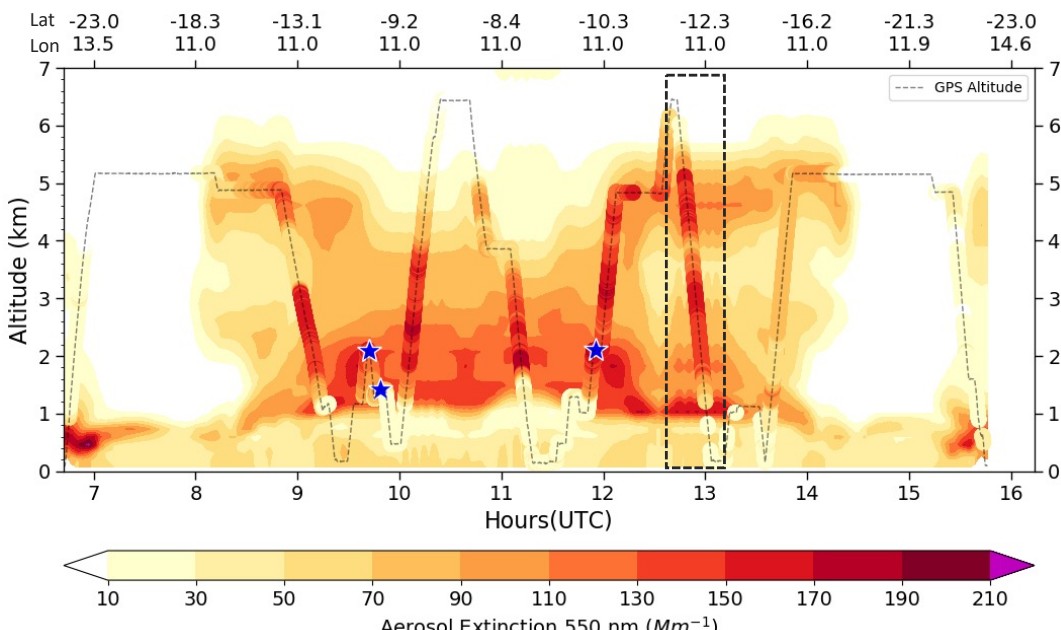

**Figure 3. Background is the 550 nm extinction profile simulated in the GEOS baseline run along the P-3 flight track on September 24, 2016. Overplotted are the P-3 measured extinction values. Dashed gray line shows the P-3 flown altitude profile along the track. Dashed black box centered at 13 UTC shows the location of the specific profile highlighted by the green star in Figure 1b and analyzed in later plots. Blue stars are the locations of specific 4STAR SSA retrievals analyzed later.**



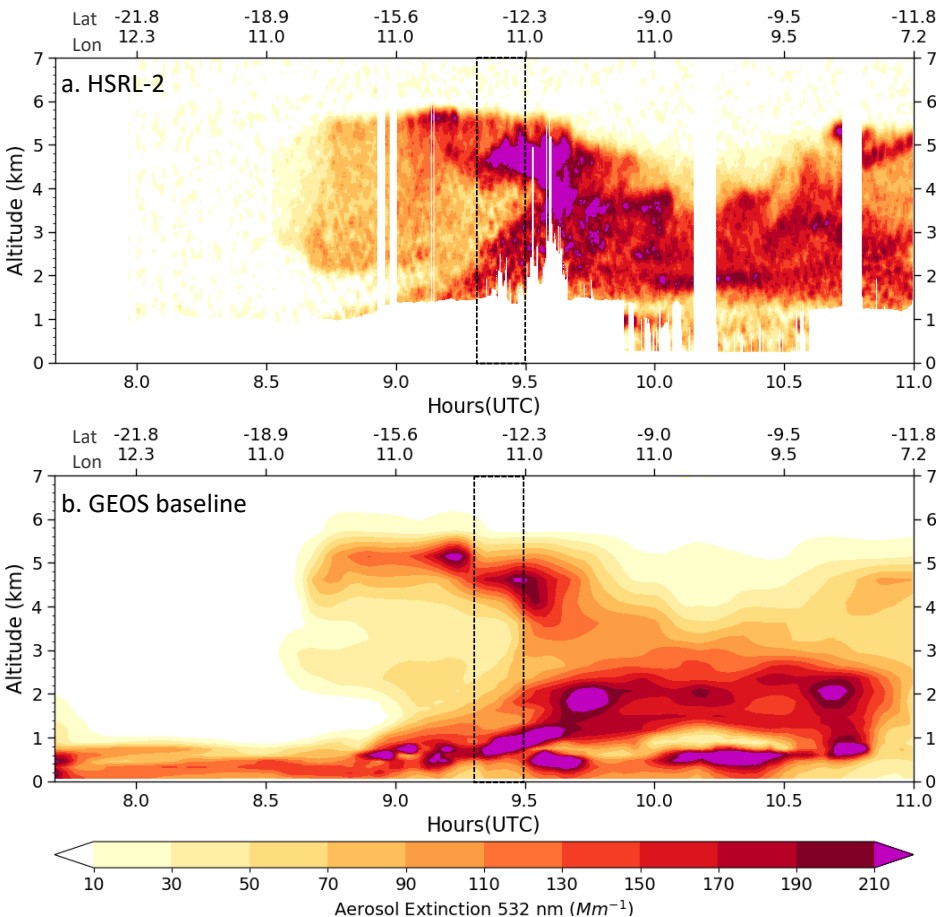


**Figure 4. (a) HSRL-2 observed 532 nm extinction profile along the ER-2 flight track on September 24, 2016. (b) GEOS baseline simulated 532 nm extinction profile along the same flight track. Dashed boxes near 9.5 UTC indicate the position of the profile compared to the P-3 profile in subsequent plots and referenced by the green star in Figure 1b.**






**Figure 5. Profiles of the P-3 in situ measured 550 nm extinction (a) and ER-2 HSRL-2 lidar 532 nm extinction (b) for the point indicated by the green star in Figure 1b. Also shown are the P-3 measured relative humidity (c) and 550 nm single scatter albedo (d). On each plot the data are indicated by the black line and the GEOS baseline simulation on the profile are indicated by the red line.**






**Figure 6. Comparison of the in-situ measured (solid lines) and GEOS baseline model simulated (dashed lines) aerosol species mass concentrations on the P-3 profile flown near 12ºS and 11ºE on September 24, 2016. Shown are (a) organic aerosol, (b) black carbon, (c) nitrate, and (d) sulfate mass concentrations.**




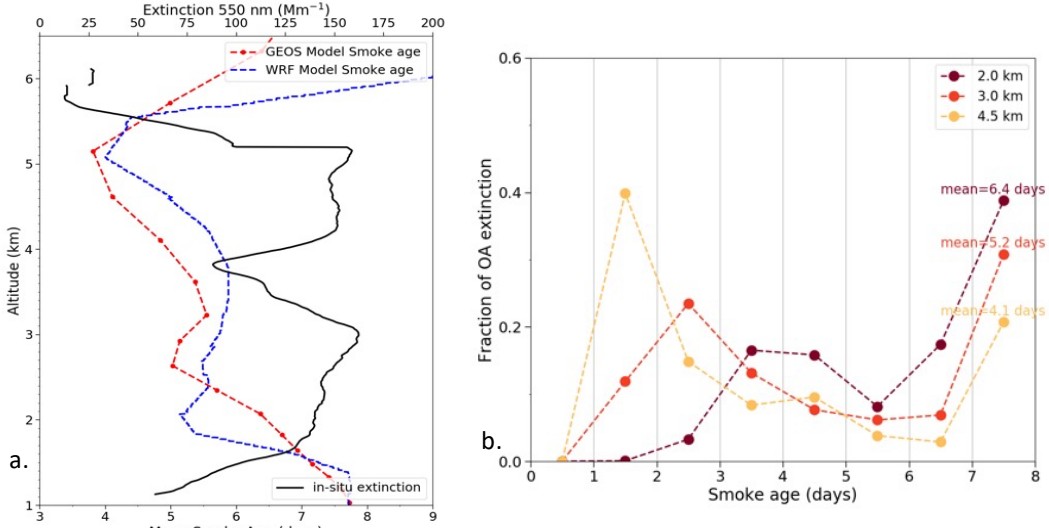

**Figure 7. (a) Comparison of the GEOS baseline simulated mean smoke age (red) to WRF-AAM simulated (blue) for the smoke profile near 12ºS and 11ºE flown by the P-3 on September 24, 2016. The in-situ extinction profile (black) is overlaid in (a) to demonstrate the differences in simulated smoke age between the multiple observed smoke layers. (b) Distribution of simulated smoke ages from the GEOS baseline simulation for the same plume profile at three different altitudes.**




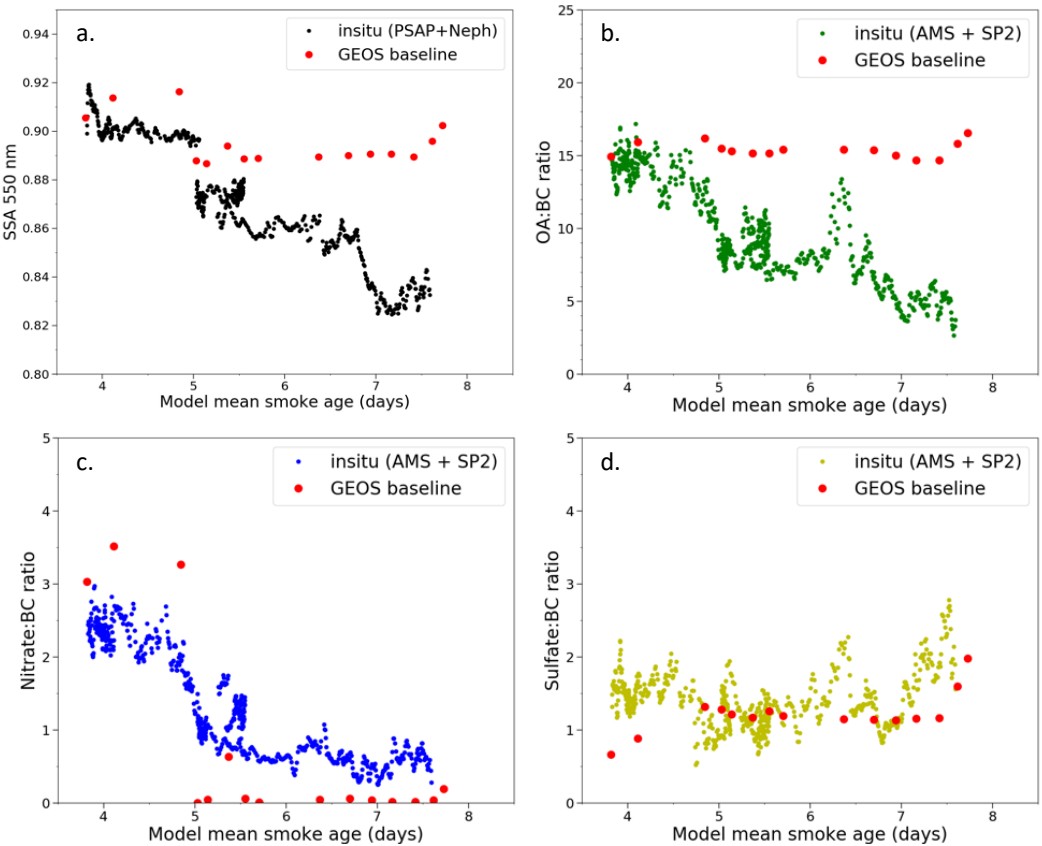

**Figure 8. Observed and simulated (a) SSA at 550 nm, (b) OA:BC ratio, (c) Nitrate:BC ratio, and (d) Sulfate:BC ratio for the smoke layers near 12ºS and 11ºE flown by the P-3 on September 24, 2016. All the results are sorted by mean smoke age from the GEOS baseline simulation.**




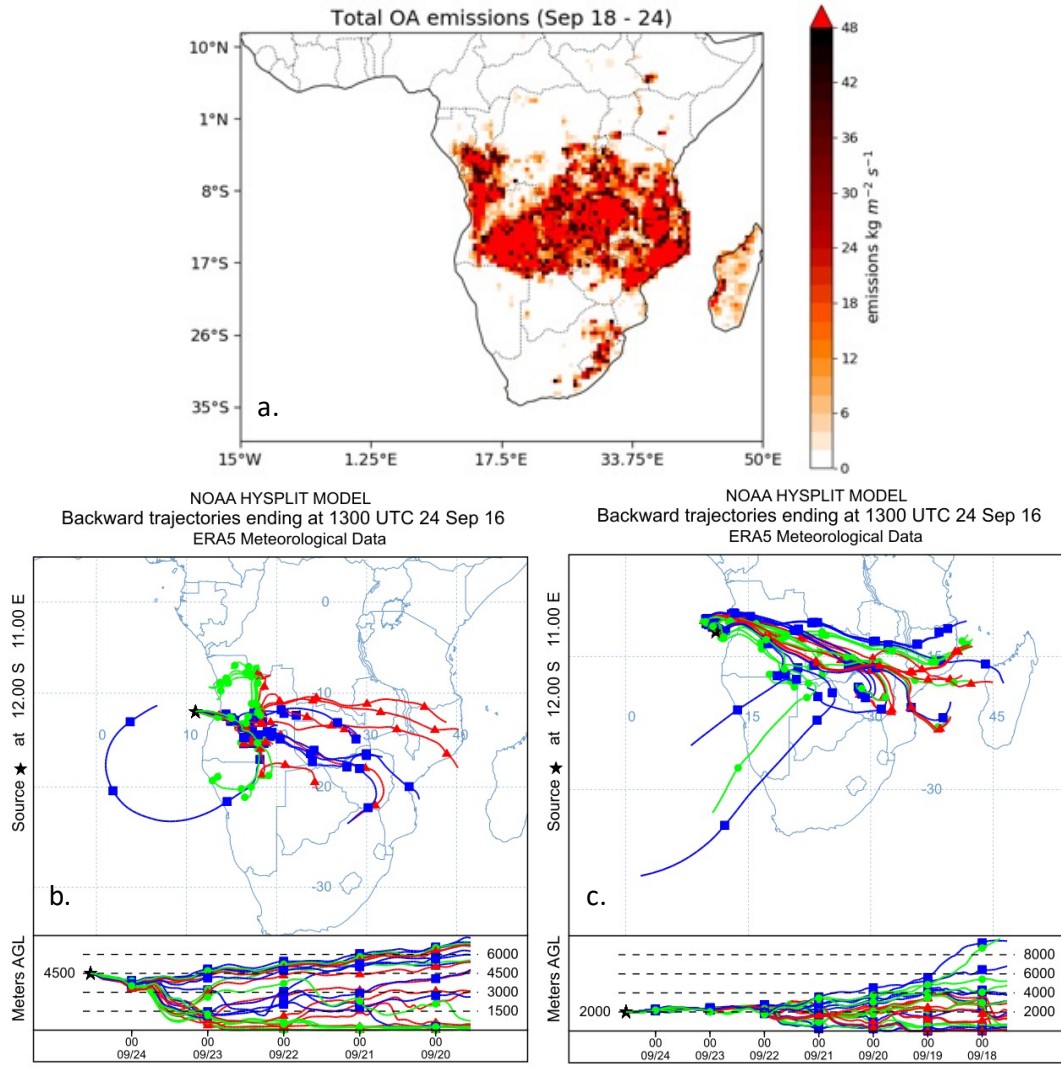

Figure 9. (a) Cumulative smoke emissions of OA over southern Africa in the week before the September 24, 2016, profile. Also shown are the ensemble of back trajectories calculated with the HYSPLIT model from the profile location initiated at 13 UTC on September 24, 2016, with trajectories ending at (b) 4.5 km and (c) 2 km altitude.





**Figure 10. (a) Fraction of the OA extinction for the September 24, 2016, profile at 4.5 and 2 km altitude attributable to emissions from various types of vegetation burned. (b) P-3 observed carbon monoxide concentrations for the September 24, 2016, profile made near 12ºS and 11ºE. (c) Ratio of excess BC to carbon monoxide for the profile. Dashed lines demarcate regimes where smoldering (ratio < 0.005) and flaming (ratio > 0.01) conditions dominate.**









**Figure 11. Model agreement across different observational spaces for Sep 24 case: GEOS model simulated (a) OA mass compared to AMS in-situ measurements, (b) dry SSA profile compared to in-situ (PSAP + Nephelometer) measurements, and (c) spectral column SSA compared to 4STAR retrievals averaged over three locations depicted by blue stars on Fig. 3.**



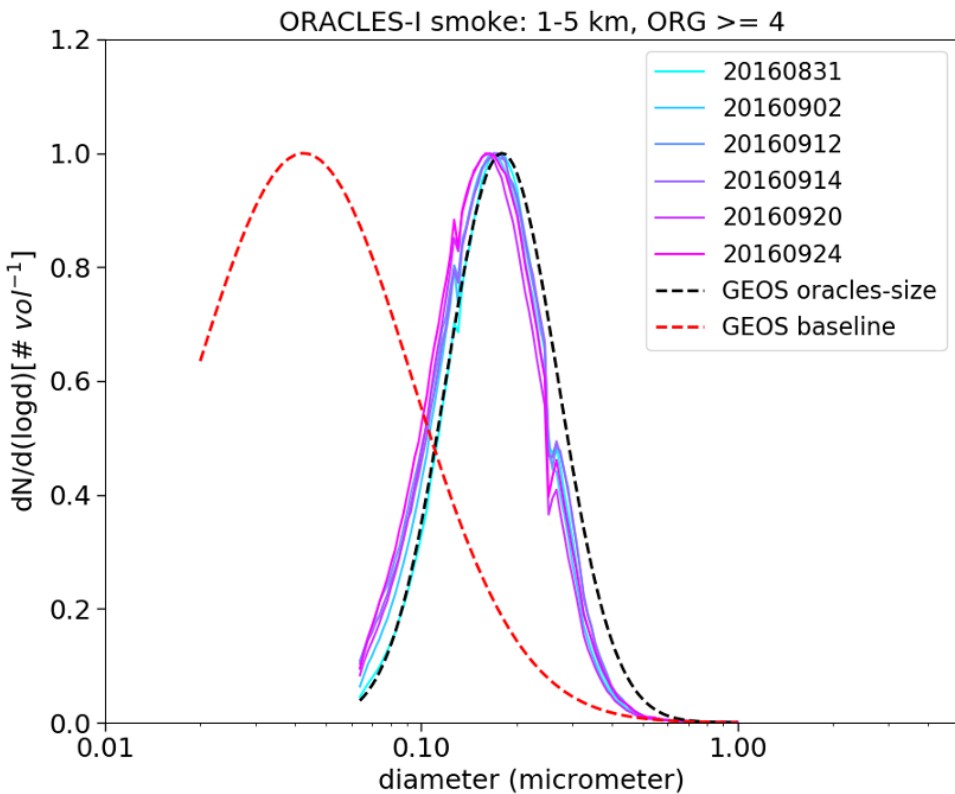


**Figure 12. Comparison of GEOS baseline and adjusted particle size distribution for organic aerosol to ORACLES observations from the UHSAS instrument.**



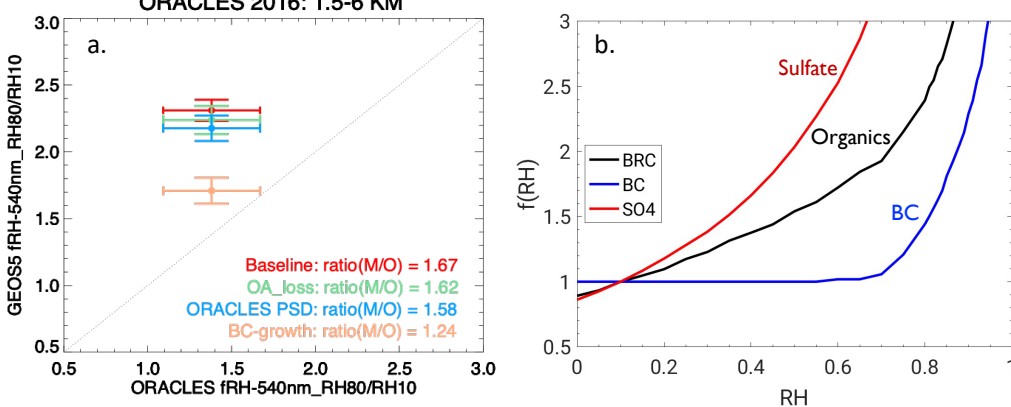

**Figure 13. (a) Comparison of simulated to observed f(RH) in smoke plumes during the ORACLES-I deployment. Different cases of the model optical property assumptions are shown, and the ratio(M/O) reports the ratio of the model to the observed mean values. (b) Default f(RH) in the GEOS lookup tables for sulfate, OA, and BC.**





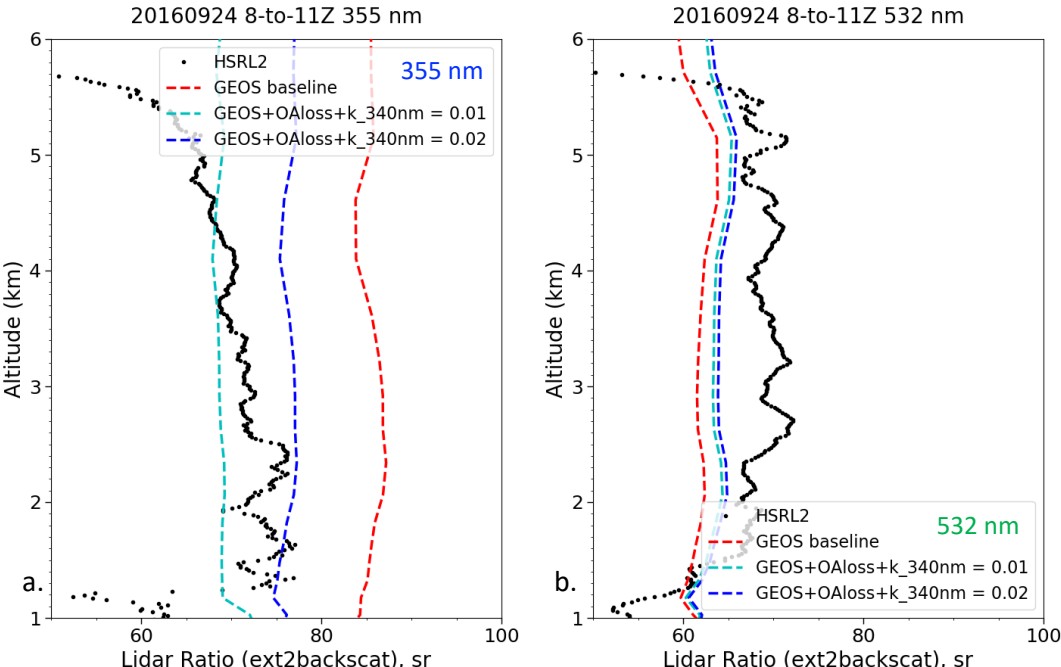

**Figure 14. Comparison of the simulated and HSRL-2 observed lidar ratios at (a) 355 nm and (b) 532 nm for the profile near 12ºS and 11ºE on September 24, 2016.**



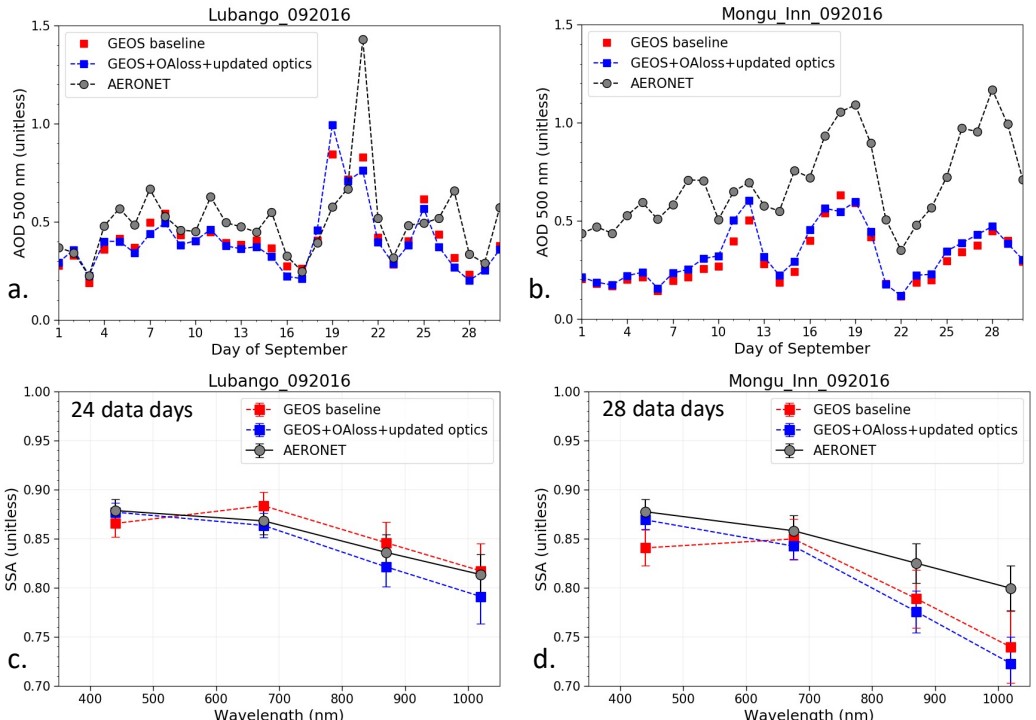

**Figure 15. Comparisons of simulated and AERONET retrieved daily AOD at (a) Lubango and (b) Mongu_Inn sites over the continent for September 2016. The corresponding monthly mean spectral SSA are also compared between model simulations and AERONET (c) over Lubango and (d) Mongu_Inn, respectively.**



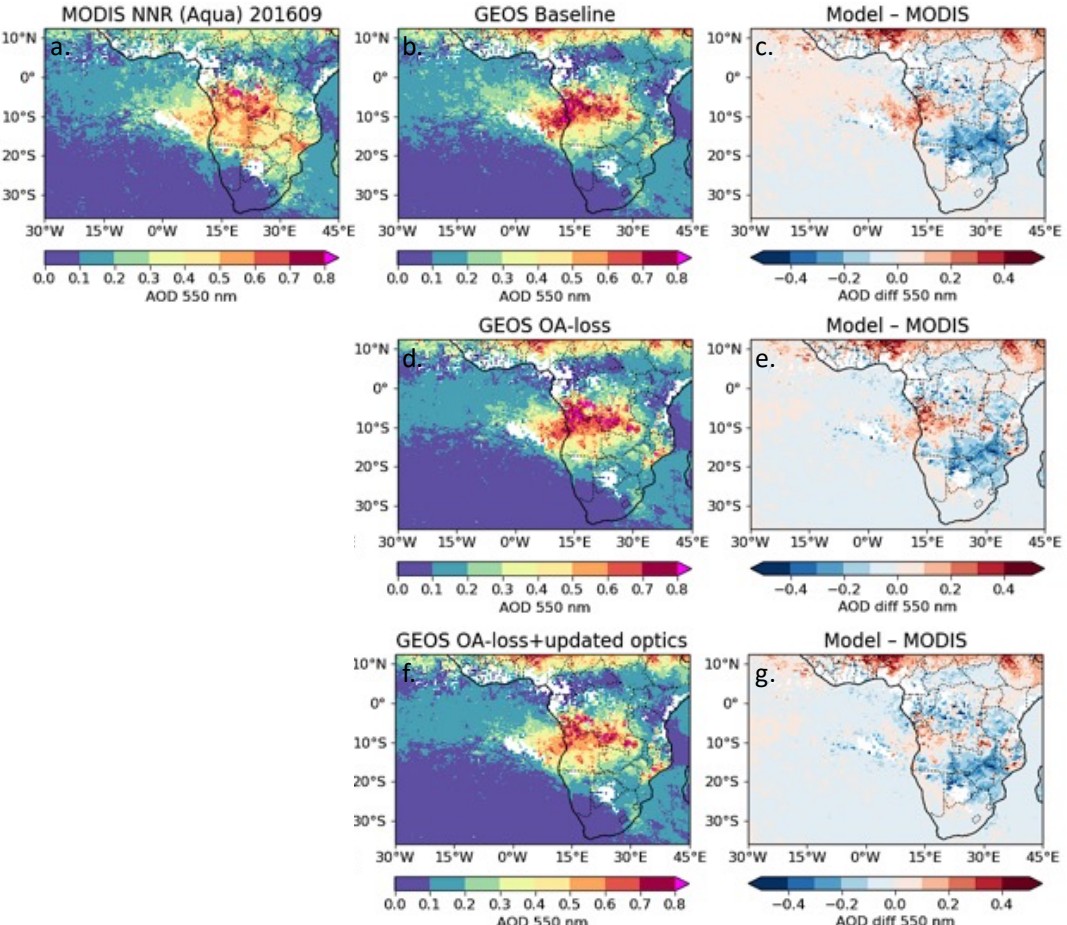

**Figure 16. Comparisons of monthly mean AOD from (a) MODIS (Aqua) NNR retrievals, (b) GEOS baseline, (d) GEOS OA-loss, and (f) GEOS OA-loss+updated optics simulations. The corresponding AOD differences between the model simulations and MODIS observations are depicted in (c, e, g), respectively.**



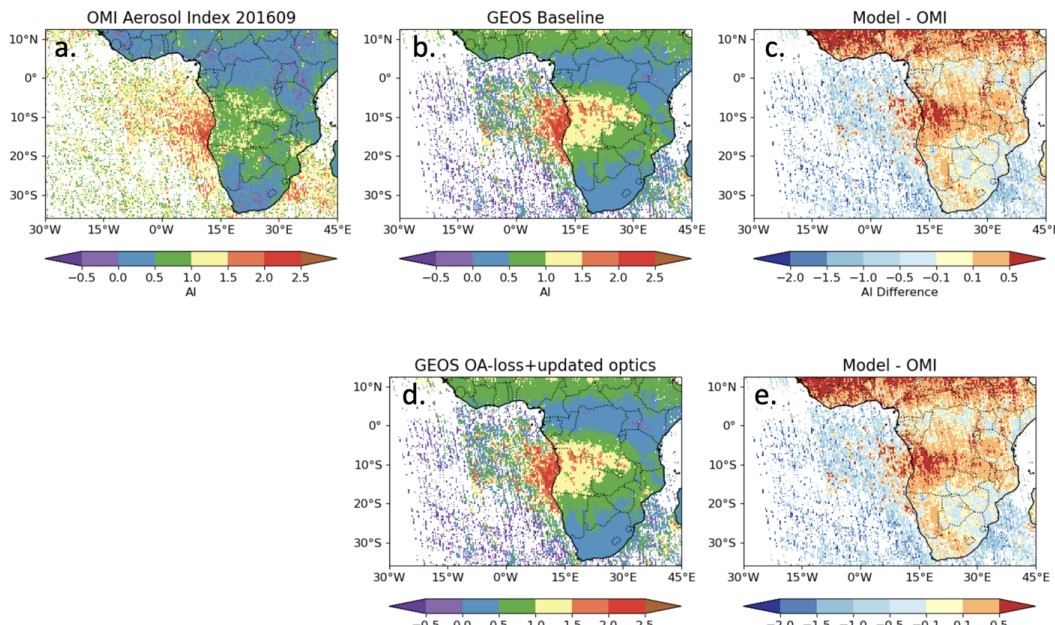

**Figure 17. The September 2016 average OMI retrieved AI (a) and the simulated GEOS AI for the baseline (b) and OAloss+updated optics (d) model runs. The GEOS – OMI difference is shown compared to the GEOS baseline run (c) and the OAloss+updated optics runs (e).**



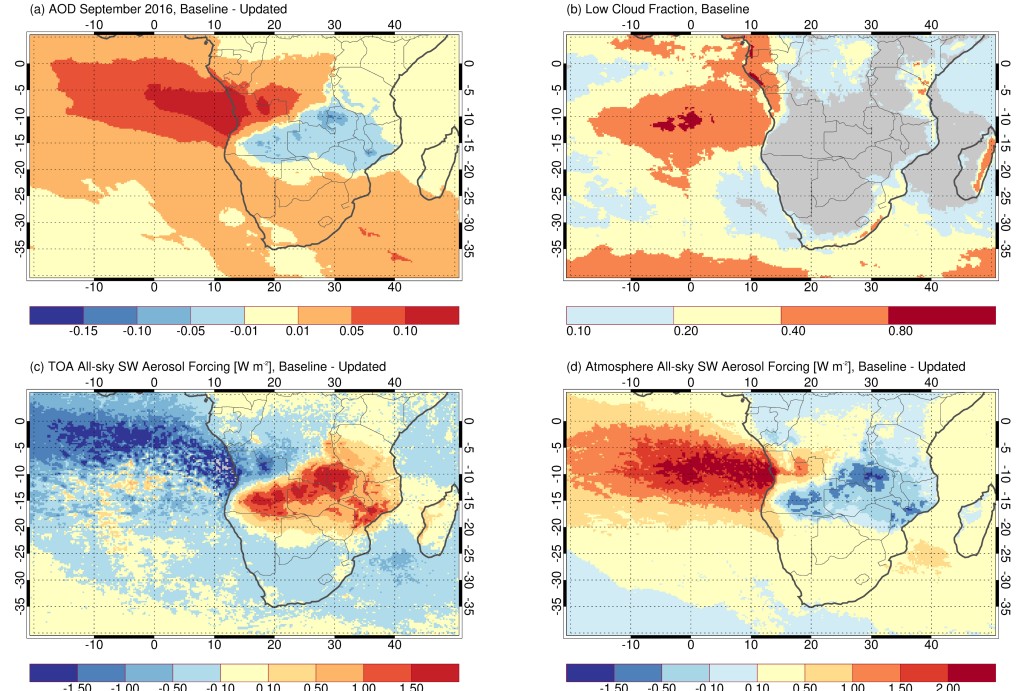

**Figure 18.** The September 2016 average GEOS-simulated (a) total 550 nm AOD difference between the baseline and OA-loss+updated optics model runs, (b) the low cloud fraction of the baseline run, (c) top-of-atmosphere all-sky shortwave aerosol forcing difference of the baseline and updated runs, and (d) the all-sky shortwave aerosol atmospheric heating difference of the baseline and updated runs. The grey area in (b) is cloud fraction less than 10%.