# Peer review of "Improved Simulations of Biomass Burning Aerosol Optical Properties and Lifetimes in the NASA GEOS Model during the ORACLES-I Campaign"

_EGUsphere, 2023_

## Referee Comment (RC1)

Das, Colarco, Bian, Gasso: Improved Simulations of Biomass Burning Aerosol Optical Properties and Lifetimes in the NASA GEOS Model during ORACLES-1 campaign.

This paper presents an improved scheme for modeling of aerosol optical properties and their time evolution, based on airborne observations made during the 2016 ORACLES field campaign over the South-East Atlantic Ocean. The title accurately reflects the scope of the work, which is withing the scope of ACP. The abstract is an accurate summary of the body of the paper.

Novel airborne data is presented from several instruments on two aircraft on one day of the campaign, and this is used to modify the aerosol optical parameterization in the GEOS model. Credit is given to the instrument teams whose results are used in the analysis. Differences between the baseline model and the improved model using several scenarios of altered aerosol optical parameterization are shown, and the radiative impacts are explored. The results are placed in a broader context by reference to AERONET time-series data and monthly-mean satellite measurements. The conclusions reached point towards a necessity for a more detailed investigation into the mechanisms driving the modified aerosol microphysical scheme developed for this study.

The authors consistently steer well clear of any sort of quantitative assessment, with a strong tendency towards the use of colloquialisms and imprecise language. Improvements in the results are frequently claimed, however reference is not often made to how the improvements are quantified by the authors. This seems like a squandering of the obviously substantial resources that were dedicated to making sophisticated field measurements, not to mention the subsequent modeling efforts performed to produce this paper. As the pre-eminent experts in the field working within a well-funded agency, it would behoove the authors to at the very least educate the reader as to what, quantitatively, they would assess to be an "improvement", a "good agreement" or a "bad agreement". Methods and assumptions made are clearly outlined. The descriptions of experiments would allow reproduction of the results by a well-funded and motivated team.

The following specific comments refer to line numbers in the pre-print:

12: "outflow region" – there are several outflow regions from the Southern African sub-continent. This study (and ORACLES) only includes one of them.

20: "mimic" – this is a rather odd anthropomorphization of the model. "model" might be more appropriate.

26: "showed a better performance" – delete "a"

44: "One of the possible reasons of the DRE mismatch" – "…for the DRE mismatch"

45: "optical properties assumptions," – "optical property assumptions"

46: "model SSAs were found to be usually higher than the aircraft" – "mostly higher…" or "frequently found to be higher…"

70: "changes in SSA along the aerosol vertical profiles" – "…within the aerosol vertical profiles"

71: "possibly with ageing smoke plumes" – "possibly with smoke plume ageing"

74: "Once we tuned…." – "After tuning…"

75: "we also utilize the larger spatial…" – "we utilize the larger spatial…"

105: "retrievals of column aerosol properties" – "retrievals of partial-column aerosol properties"

106: "under certain flight conditions" – "under ideal flight and atmospheric conditions…"

107: "due to suspected stray light contamination" – this may also be due to an incorrect retrieval of the column NO2, leading to a misattribution of observed total extinction to aerosols.

115: "above-column SSA" – it's not clear what this is, do you mean "above-aircraft column SSA".

116: "… was used to emphasize on SSA of smoke layers" – " to focus on SSA of smoke layers"

123: "we mainly use the aerosol extinction and lidar ratio…" – it's not clear what you mean. Did you, or did you not, use other data from the HSRL?

126: "interpolated to the aircraft GPS times" - it's not clear what this means.

128: "… MBL which is mostly capped" – "frequently capped/always capped/sometimes capped". "Mostly" is quite meaningless in this context. What is the significance of MBL stratocumulus clouds here? Do they have an impact on the space-based lidars that is not a factor for the ER-2 HSRL?

136: "we convert them to ambient conditions…" is this done by the team writing this paper, or by the HIGEAR team?

140: where was the UHSAS mounted on the aircraft? Which aircraft?

143: Where was COMA mounted on the aircraft? Which aircraft?

169: The way you phrase it here, it seems like the row anomaly is an additional consideration, but you don't say what you have done to mitigate it.

~175: Why is the NNR better than the MODIS C6.1 data for your work? It's not clear why this is the better choice, without the reader going and reading and comparing Levy 2013 ad Randles 2017.

181: "The target of the NNR algorithm is the log-transformed AERONET…" – it's really not clear what this means. AERONET provides effectively a point measurement, yet NNR provides a 10km spatial resolution product. Is AERONET used as training data for the NNR? Clearly, saying that the NNR provides AERONET AOD is nonsensical, since these are different instruments.

198: Does DMS have a wind-blown source? Sea-salt production may be related to windspeed, but is DMS not biogenic? How does the windspeed influence ocean productivity? Please clarify and provide a reference, if this is indeed how it is represented in the model.

209: "… following (Kim et al, 2015)" – "following Kim et al (2015)"

210: "conversion of VOC to SOA a simple function…"  - "…modeled using a simple function…"

215: "aerosol species are externally mixed…" – this is a very odd sentence construction. "…species are considered to be externally mixed…"

232: is Collow et al still in preparation?

233: "see also Das et al 2017)" – missing parenthesis.

239: "(Table 1)" – perhaps something like "(compared with the other simulations in Table 1)"

240: "day of the week its emissions…" – "day fo the week on which its emissions were injected"

243: It's too late to change this study now, but surely using a non-perishable tracer such as Julian day would be better?

265: "closer together is to the translate" – delete "the"

275: "After the Colarco et al study…" – "Since the study by Colarco et al…"

282: "almost the same" – how do you quantify this?

283-4: "See Colarco et al for a detailed description of the AI simulator" – no need to say the same thing twice…

286: "Linux-distribution" – is the hyphen necessary/correct? Your editor may have an opinion.

286-290: Three sentences in a row start with "We use…"

293: It's not clear how these offsets are done. Is the entire meteorological field shifted while keeping the initial position the same? Is there some reference describing this in detail?

294: "all-possible" – hyphenation is dubious.

299: "Sept 2016 in context of…." – "in the context of…"

302: The aircraft did not observe anything, they just fly through the air. The instruments do the observing (or perhaps strictly speaking, not even that; however this is more a question of philosophy) Perhaps: "…observed from both the P-3 and ER-2…"

307: "We suggest the vertical variation of SSA…." This seems completely out of context here. Is this one of your hypotheses?

310: "vertical flight trajectory" – "vertical flight profile". "Trajectory" is strictly speaking something that is followed by e.g. artillery shells or re-entry vehicles. In the horizontal plane, the aircraft would follow a "track".

311: What is the time difference between the airborne measurement and the GEOS model time-step?

313: "… with part of P-3 flight path" – "with part of the P-3 flight track".

316 & Fig 3 & Fig 4. The P-3 appears to have been flying northbound until approx. 11:30UTC, then turning southbound. The ER-2 appears to have been flying northbound until around 10.1 UTC. Thus, these plots are folded back on themselves in latitude. This is quite confusing, and a clearer description of these plots is warranted.

317: Refer to the specific plot (i.e. Fig 3?). "About 13UTC" is very imprecise; the profile starts well before 13UTC. Please furnish the exact times that you are referring to. Similarly: "About 6km to 1km" – please furnish the exact altitudes that you are referring to. Also, improve the horizontal scale markings on the plot so that the reader can follow these exact times that you furnish.

318: (~9.5 UTC) – please furnish the exact time. You highlight a box prior to 9.5 UTC on the plot. Is this the time you refer to, or is it the high-extinction plume between 3.2km and 5.2km altitude, shortly after this time that you wish to draw the reader's attention to?

322: "extinction magnitudes can be explained in part…" – This is a rather qualitative assessment, with a equally vague description. Can you provide a quantitative assessment of this?

325: "at least for the upper level smoke layer" – "upper level smoke layer" might be subject to interpretation. Starting off your analysis of the profiles with a precise description of what you see (e.g. an upper smoke layer measured by the ??? instrument in the P-3 aircraft while descending between 4.2 – 5.2 km and a lower smoke layer at 1.6- 3.6km…) would get you and your reader on the same page and allow you to clarify a lot of the vague language in these  and following paragraphs.

325: "model simulated" – your editor might want a hyphen here, since "model-simulated" is an adjective, although Copernicus language editors seem to ignore this common feature of English grammar.

Fig 5c: have you tried plotting temperature on this plot? The existence of absolutely stable layers in the winter atmosphere over the sub-continent is a feature that was described decades ago by Tyson. It would be interesting to see that they exist over the ocean too.

326: "having a very good match of simulated RH" – delete "having"

339: "underestimated in the model overall" – by what measure? can you quantify this?

339-340: "multiplume" – this is not an established term, and has all sorts of connotations which will differ from one reader to another. Perhaps "layered" is better?

341: "Here, BC is the primary…" are you referring to a specific figure, or a specific profile, or a specific model simulation? This is not clear.

343: Same comment. Are you referring to a specific figure?

346: "(Table 1, Section 2.4.1)" – it would be polite so say "(described in Table 1 and Section 2.4.1)" rather than barking at your reader.

346: "almost monotonic" is meaningless. It's either monotonic or not, and it seems from the plot that is is clearly not. What do you mean?

351: "demonstrating that the models are in close agreement" – by what metric?

352 & Fig 7b. Would it not be clearer to use something like a box-and-whisker plot on the vertical profile. How does the distribution of extinction fraction in the "clean" layer at 3.8km look?

359: "…causing the weighted mean smoke age to be younger than it possibly is" could this be rephrased to make it clearer?

362: "… composition of aerosols change with smoke age" – "changes"

366: If these quantities are correlated, could they be plotted against each other? This would make your point clear, rather than the qualitative "by inspection" correlation that you claim.

369: "nitrate:BC and SO4:BC" – consistent nomenclature: choose words or formulae

374: This introduces your hypotheses, however this sentence is quite weak, almost an afterthought.

377: "with perhaps different…" – this is your hypothesis. "Perhaps" makes it weak and negotiable. State it clearly and boldly.

380: "origin locations" – delete locations.

385: It's not clear how Fig 9a is arrived at. Is this from QFED introduced in line 216, then never mentioned again?

390-392: "further suggest that the contribution… are possibly causing the…" – "suggest" + "possibly" in one sentence makes for overwhelming uncertainty. What do you want to say here?

393: "suggest… almost…" here makes me think that you have no faith in this tool whatsoever. Why do you use it?

396: "as a proxy for vegetation type". Surely the OA:BC is *determined* by vegetation type?

399: Fig 10 is introduced rather suddenly here, and the reader is left to figure out its meaning with no guidance from the author.

407: "do not even differentiate" – delete "even"

409: "the most different fuel types" – it's not clear what is meant by this.

410: savannas: check for consistent spelling cf. line 422.

Table 2: where does the multiplier of 1.8 come from? Is this an empirically established ratio? Is there a reference for this?

417: "emission ratios based to fire characteristics…" – "based on fire char…"

422: spelling of savannah, cf. line 410.

422: "savannah and grassland region" – plural "regions"

424: "outside plume (>6km) values" – "values outside the plume (found at altitudes > 6km)"

427: "Therefore…" here you refute your hypothesis. I would consider having this sentence stand as a paragraph on its own.

442: "increase in particulate oxygen" – it's not at all clear what this is. Is this a change in oxidation state? Are there unbound oxygen atoms resulting from a chemical reaction? Are particles being oxidized?

448: "prescribed at emissions…" – at the point of emission?

448: "the burning fuel type" – delete "burning"

452: "further aged smoke" – in plain language: "older smoke"

462: "brought the modeled SSA curve closer to 4STAR" – by what metric? Some wavelengths look "close", some wavelengths look "far". What are the implications of this? If you are giving a qualitative assessment, base it on some numbers.

470: you could refer to Fig 10 here, if I understand it correctly.

485 & Fig 13: f(RH) nomenclature in Fig 13 axis labels is confusing and unclear.

486: f(RH) is high/low, whereas here you mention low followed by high. This is confusing.

489: here you use measurements between 1.5 – 6km. On line 479 you use 1.0 – 5km ostensibly to achieve the same aim.

496: "Sulfate hygroscopicity increases…" do you mean "hygroscopic growth factor"? These are not the same thing.

501: "tracks with BC" – this is an extreme colloquialism. Meaningless. What are you trying to say?

501-2: "closest match to observations"  - can you provide some numbers that you used to guide your assessment?

506: "blue (or 355nm) channel"  -  is it blue, or is it 355nm? or is it blue and 355nm?

507: "by (Veleovskii et al, 2020) : V. et al (2020)

510: "were set" – past tense? "was set…"

511: Give the precise definition and be done with it. If there is a need for a looser definition, provide this afterwards.

Fig 14 & lines 510, 517, 519, 520: is it k_340 or k_350? The figure and the text do not correspond.

521: Why is it worth mentioning this? How does this impact your study? Perhaps a better phrasing would be "it is worth mentioning that GOCART assumes particles to be spherical, notwithstanding Meng et al (2010)…"

525: "we call as 'updated optics'…" – delete "as"

527: "these optics update" – "this optics update"

528: "slight decrease" – what constitutes a slight decrease?

529: "very close" – what constitutes very close?

533: "assumptions that is leading to.." – "that are leading to"

534: Explian by what metric it shows a "better match"? You have some measurements here that took incredible resources to make, yet you provide a qualitative description.

541: "the September" – delete "the"

543: does not deviate much. Add some numbers to your qualitative assessment. These measurements took incredible resources to make.

546-7: "suggests that model is missing" – insert "the"

552: "over complete SE Atlantic" – "the broader SE Atlantic region" or something like that.

556: "the Model and MODIS NNR AOD" – it would be worthwhile to remind the reader here how the model AOD is derived.

557: "Overall, there is a good match…" – by what metric?

558: "…compared to the source region…" it's not clear what this means. Is the "overall good match" better over the ocean than over the source region? By what metric?

Perhaps a plot of AOD from MODIS and the model runs along the 10deg S latitude line would illustrate the point you are trying to make more clearly.

575: you speak of retaining OMI pixels without cloud contamination, in the context of the model-based AI calculator. This is clearly meaningless; the discussion about cloud contamination should come after you introduce the OMI observations in line 576.

585: I can see hardly any difference between Fig 17c and Fig 17e. How do you quantify the "much more favorable comparison"?

591: The direct radiative forcing of the aerosols is what it is in the real world out there over the SE Atlantic. You are talking about the modeled DRF.

598-600: "due to aerosol difference between our two runs… the other without the aerosols" – do you mean without aerosols, or do you mean the difference between the baseline model and the modified/updated aerosol scheme?

603: this is the first mention in this paper of the "main cloud feature". What is this cloud feature?

606: "reflected radiation to space" – "radiation reflected to space"

607: it's not clear where the border of DRC and Zambia is. What lon/lat are you referring to.

613: You introduce the cloud features in the last line of your section. What clouds are you referring to? Modeling of cloud features has not been mentioned prior to this.

621: "move observing smoke plume" – do you mean "absorbing"?

622: "We hypothesize a loss process … explains…"  - "We hypothesize a loss process … that explains" or "loss process… explaining"

624: "mimic" is again an odd anthropomorphizing of the model.

634: "a better performance…" issues of how you quantify this have been addressed in the sections above.

639: consider deleting "In terms of future directions" and start the sentence with "The simplistic approach…"

643: "mimic"

---

## Referee Comment (RC2)

**Comments for: Improved Simulations of Biomass Burning Aerosol Optical Properties and Lifetimes in the NASA GEOS Model during the ORACLES-I Campaign by Sampa Das et al. (2023)**

This work evaluated NASA GEOS model's ability to simulate biomass aerosol properties by comparing with observations from field campaign ORACLELS-I. Sampa Das et al. implemented two adjustments in GEOS model to improve organic aerosol aging process and optics caculation. These two adjustments were further evaluated by AERONET and satellite observations. Radiative implication of these adjustments were discussed in the end. The content aligns well with the *ATMOS CHEM PHYS*'s scope and I recommend considering publication after major revisions.

Major comments:

- The abstract is almost identical with the first paragraph of conclusion. Abstract can be less detailed in methods, while the conclusion can provide more information about the adjustments implemented in the model.

- There are substantial text, figure and citation format issues in the draft. Details will be listed in the Minor comments section.

- For the first adjustment, are the 60% increased OA and 15% increased BC emission consistent with previous studies, or just to compensate the OA loss process introduced in this work? For the second adjustment about hygroscopic growth, it is not too clear and please add more descriptions about this adjustment.

- From the spatial distribution differences in figure 16 and 17, it is not clear whether the models with the implemented adjustments are more consistent with the observations or not. Please better quantify the differences by some metrics. Either regional mean differences or spatial correlation will work.

Minor comments:

For the improper text formats:

- Certain acronyms are repeated or mentioned more than once, such as AOD in line 175 and line 105, TOA in line 37 , line 178 and line 598. Please check other acronyms throughout the papers.

- There should be spaces between the math operation symbols and the numbers, such as line 100: $<40\%$, should be $< 40\%$. Also the equal operations in line 629 and line 632. Please check this issue throughout the papers as well.

For the improper citation formats:

- Citations for data sets including links should be corrected: line 97 and line 253.

- Citations formats when abbreviations are included are wrong: line 119 for HSRL-2, line 156 for OMI.

- In-text citation formats are incorrect.
  –Line 108: Dubovik and King, (2000), no comma in the middle.
  –Line 214: Colarco et al., (2014), no comma in the middle.

For the improper figure formats:

- Figure title formats are inconsistent between the figures, neither the font nor the sizes. Please keep the title formats consistent of all the figures.

- The units in the axis caption should not be italic, such as $Mm^{-1}$ in figure 4, should in M m$^{-1}$. Please check the unit formats in other figures and change properly.

- The titles of some subplots in the same figure are the same and unnecessary, such as figure 6(a)-(d), figure 9(b)-(c), figure 10(b)-(c), figure 11(a)-(b). You can either delete the title, or use other more informative texts.

- Subplots in figure 15 have the same legend captions. If the legends are the same for all the subplots, no need to show them in all subplots.

- The label notation (a)(b)(c)(d) is very inconsistent between the figures, sometimes outside the figure, such as figure 1, sometimes inside the figure, such as figure 4. Also, the labels sometimes are not clear if overlapping over other colors, such as figure 16. Please try to keep the position of these labels consistent among the figures and keep them clear.

- The names of the cases are inconsistent between the figures. For the OA loss adjustment cases, the notation is +OAloss_6days in figure 11, while it is GEOS + OAloss in figure 14 and GEOS OA-loss in figure 16. It is better to describe all the simulations in table 1. Create the cases names that are easier to put in the figures and be consistent among the figures.

- The legends in figure 13(b) are inconsistent with the text notations along the lines. Since they describe the same thing, you can delete either one.

---

## Author Response (AR1)

Response to Reviewer Comments on "Improved Simulations of Biomass Burning Aerosol Optical Properties and Lifetimes in the NASA GEOS Model during the ORACLES-I Campaign" by Das et al.

We thank the reviewers for their carefully considered and detailed review of the paper. Following are authors' responses in red and original comments of the reviewers are in black.

Reviewer #1

This paper presents an improved scheme for modeling of aerosol optical properties and their time evolution, based on airborne observations made during the 2016 ORACLES field campaign over the South-East Atlantic Ocean. The title accurately reflects the scope of the work, which is withing the scope of ACP. The abstract is an accurate summary of the body of the paper.

Novel airborne data is presented from several instruments on two aircraft on one day of the campaign, and this is used to modify the aerosol optical parameterization in the GEOS model. Credit is given to the instrument teams whose results are used in the analysis. Differences between the baseline model and the improved model using several scenarios of altered aerosol optical parameterization are shown, and the radiative impacts are explored. The results are placed in a broader context by reference to AERONET time-series data and monthly-mean satellite measurements. The conclusions reached point towards a necessity for a more detailed investigation into the mechanisms driving the modified aerosol microphysical scheme developed for this study.

The authors consistently steer well clear of any sort of quantitative assessment, with a strong tendency towards the use of colloquialisms and imprecise language. Improvements in the results are frequently claimed, however reference is not often made to how the improvements are quantified by the authors. This seems like a squandering of the obviously substantial resources that were dedicated to making sophisticated field measurements, not to mention the subsequent modeling efforts performed to produce this paper. As the pre-eminent experts in the field working within a well-funded agency, it would behoove the authors to at the very least educate the reader as to what, quantitatively, they would assess to be an "improvement", a "good agreement" or a "bad agreement". Methods and assumptions made are clearly outlined. The descriptions of experiments would allow reproduction of the results by a well- funded and motivated team.

We acknowledge the reviewer's substantive concern on the lack of precise language to quantify the improvement we sought to make in the model and have attempted to address the issue both in our responses to specific points below and in the general tone of the paper. In some cases, we responded to multiple comments of the reviewer with a single response that reflected substantially modified text; these "grouped" responses have been indicated by highlighting the relevant comments.

The following specific comments refer to line numbers in the pre-print:

12: "outflow region" – there are several outflow regions from the Southern African sub-continent. This study (and ORACLES) only includes one of them.

Restated as: "In order to improve aerosol representation in the NASA Goddard Earth Observing System (GEOS) model, we evaluated simulations of the transport and properties of aerosols from southern African biomass burning sources that were observed during the first deployment of the NASA ORACLES (ObseRvations of Aerosols above CLouds and their intEractionS) field campaign in September 2016."

20: "mimic" – this is a rather odd anthropomorphization of the model. "model" might be more appropriate.

Replaced with "simulate"

26: "showed a better performance" – delete "a"

Corrected

44: "One of the possible reasons of the DRE mismatch" – "...for the DRE mismatch"

Corrected

45: "optical properties assumptions," – "optical property assumptions"

Corrected

46: "model SSAs were found to be usually higher than the aircraft" – "mostly higher..." or "frequently found to be higher..."

Restated as "were frequently found to be higher than"

70: "changes in SSA along the aerosol vertical profiles" – "...within the aerosol vertical profiles"
71: "possibly with ageing smoke plumes" – "possibly with smoke plume ageing"

Restated as: "ORACLES-I observations showed variability in the vertical profile of the smoke SSA on several flights, possibly related to smoke plume aging"

74: "Once we tuned...." – "After tuning..."

Corrected

75: "we also utilize the larger spatial..." – "we utilize the larger spatial..."

Corrected

105: "retrievals of column aerosol properties" – "retrievals of partial-column aerosol properties"

Corrected

106: "under certain flight conditions" – "under ideal flight and atmospheric conditions..."

Corrected

107: "due to suspected stray light contamination" – this may also be due to an incorrect retrieval of the column NO2, leading to a misattribution of observed total extinction to aerosols.

We checked with the 4-STAR team, and they maintain stray light is the main culprit. We have added the appropriate citation: "However, due to suspected stray light contamination within the 4STAR spectrometer around 440 nm (Pistone et al., 2019), the set of input wavelengths for 4STAR were modified to be 400, 500, 675, 870, and 995 nm (as opposed to the standard AERONET input wavelengths)."

115: "above-column SSA" – it's not clear what this is, do you mean "above-aircraft column SSA". 116: "... was used to emphasize on SSA of smoke layers" – " to focus on SSA of smoke layers"

Restated as: "Since 4STAR retrieves the above-aircraft column SSA, our last screening criterion was used to focus on the SSA of smoke layers above the boundary layer and exclude the influence of marine aerosols within the boundary layer. "

123: "we mainly use the aerosol extinction and lidar ratio..." – it's not clear what you mean. Did you, or did you not, use other data from the HSRL?

Removed the word "mainly"

126: "interpolated to the aircraft GPS times" - it's not clear what this means.

We have clarified the text: "Out of the standard aerosol data products (Burton et al., 2012) that HSRL-2 provides, we use the aerosol extinction and lidar ratio (extinction-to-backscatter ratio) profiles at 355 and 532 nm for this study, both provided in 60 second averages to improve the instrument signal-to-noise (equates to ~12 km along-track horizontal resolution of the aircraft)."

128: "... MBL which is mostly capped" – "frequently capped/always capped/sometimes capped". "Mostly" is quite meaningless in this context. What is the significance of MBL stratocumulus clouds here? Do they have an impact on the space-based lidars that is not a factor for the ER-2 HSRL?

We have removed this confusing sentence.

136: "we convert them to ambient conditions..." is this done by the team writing this paper, or by the HIGEAR team?

We did this conversion because the model is presented at ambient conditions. We have restated as "The AMS-measured mass concentrations are provided at standard temperature (273 K) and pressure (1000 hPa), but we convert them here to ambient conditions using the ideal gas law and measured pressure and temperature information before comparing with the model equivalents."

140: where was the UHSAS mounted on the aircraft? Which aircraft? 143: Where was COMA mounted on the aircraft? Which aircraft?

We have included the following text: **"Aerosol Size Distribution:** Particle size distributions used in this study were measured from the P-3 with an ultra-high-sensitivity aerosol spectrometer (UHSAS, Droplet Measurement Technologies, Boulder CO, USA) with a fuselage-mounted inlet. UHSAS is an optical-scattering, laser-based aerosol particle spectrometer that measures particles from 60–1000nm at 1s time resolution, thereby covering the entire accumulation mode. The UHSAS measured size distribution is reported as particle number concentrations (in $cm^{-3}$) per size bins that are approximately logarithmically spaced.

**Carbon Monoxide (CO):** CO was measured from the P-3 with a gas-phase CO/CO2/H2O analyzer (ABB/Los Gatos Research $CO/CO_2/H_2O$ analyzer known as COMA) with an inlet mounted on the aircraft fuselage. It uses off-axis integrated cavity output spectroscopy (ICOS) technology to make stable cavity enhanced absorption measurements of CO, $CO_2$, and $H_2O$ in the infrared spectral region (Provencal et al., 2005). The measurements were reported as dry air volume mixing ratios in parts per billion (ppbv)."

169: The way you phrase it here, it seems like the row anomaly is an additional consideration, but you don't say what you have done to mitigate it.

Unfortunately, we cannot mitigate the satellite instrument error. We have restated as: "The OMI swath ideally provides near-daily global coverage but has been impacted by a "row anomaly" defect since shortly after launch that has effectively degraded its coverage by about 50% so that OMI now achieves global coverage every two days (Torres et al. 2018). In our comparisons that follow we sample model output only where valid OMI data are collected."

~175: Why is the NNR better than the MODIS C6.1 data for your work? It's not clear why this is the better choice, without the reader going and reading and comparing Levy 2013 ad Randles 2017.

We have attempted to clarify the discussion: "However, instead of directly using the MODIS operational retrievals of aerosol optical depth (AOD) for model evaluation, we use a bias-corrected AOD dataset, called the MODIS NNR, which was derived initially for use in the Modern-Era Retrospective analysis for Research and Applications, version 2 (MERRA-2, Randles et al., 2017) aerosol reanalysis. As much care as is taken in creating the MODIS standard products, there

are nevertheless significant biases related to cloud and land features that must be screened prior to using these data in assimilation systems (e.g., Zhang and Reid, 2006). The NNR refers to a Neural Net Retrieval algorithm that computes AERONET-calibrated AOD from satellite-based radiances, in this case the same MODIS collection 6.1 radiances used in the standard retrieval products."

Randles, C. A., da Silva, A. M., Buchard, V., Colarco, P. R., Darmenov, A., Govindaraju, R., Smirnov, A., Holben, B., Ferrare, R., Hair, J., Shinozuka, Y., and Flynn, C. J.: The MERRA-2 Aerosol Reanalysis, 1980 Onward. Part I: System Description and Data Assimilation Evaluation, J Clim, 30, 6823–6850, https://doi.org/https://doi.org/10.1175/JCLI-D-16-0609.1, 2017.

Zhang, J. and Reid, J. S.: MODIS aerosol product analysis for data assimilation: Assessment of over-ocean level 2 aerosol optical thickness retrievals, J Geophys Res Atmospheres 1984 2012, 111, D22207, https://doi.org/10.1029/2005jd006898, 2006.

181: "The target of the NNR algorithm is the log-transformed AERONET..." – it's really not clear what this means. AERONET provides effectively a point measurement, yet NNR provides a 10km spatial resolution product. Is AERONET used as training data for the NNR? Clearly, saying that the NNR provides AERONET AOD is nonsensical, since these are different instruments.

We have attempted to clarify this discussion: "The NNR algorithm is trained on the log-transformed AERONET AOD interpolated to 550 nm and co-located with MODIS observations of the predictors. Application of the NNR to the MODIS products is found to produce a higher quality AOD product compared to (relatively unbiased) AERONET observations (Randles et al., 2017)."

198: Does DMS have a wind-blown source? Sea-salt production may be related to windspeed, but is DMS not biogenic? How does the windspeed influence ocean productivity? Please clarify and provide a reference, if this is indeed how it is represented in the model.

We have attempted to clarify this discussion: "Bulk sulfate mass is tracked, with primary emissions from anthropogenic sources and precursor emissions of dimethylsulfide (DMS), which has a wind-blown source function over the ocean scaled to observed DMS surface concentrations (Lana et al. 2011), and sulfur dioxide ($SO_2$), which has emissions from anthropogenic, volcanic, and biomass burning sources."

Lana, A., Bell, T. G., Simó, R., Vallina, S. M., Ballabrera-Poy, J., Kettle, A. J., Dachs, J., Bopp, L., Saltzman, E. S., Stefels, J., Johnson, J. E., and Liss, P. S.: An updated climatology of surface dimethlysulfide concentrations and emission fluxes in the global ocean, Global Biogeochem Cy, 25, n/a-n/a, https://doi.org/10.1029/2010gb003850, 2011.

209: "... following (Kim et al, 2015)" – "following Kim et al (2015)"

Corrected

210: "conversion of VOC to SOA a simple function…" - "…modeled using a simple function…"

Corrected

215: "aerosol species are externally mixed…" – this is a very odd sentence construction. "…species are considered to be externally mixed…"

Restated as: "The aerosol species are externally mixed in the model for optics and chemistry purposes."

232: is Collow et al still in preparation?

Restated as "not presented here." The paper is still in preparation.

233: "see also Das et al 2017)" – missing parenthesis.

Corrected

239: "(Table 1)" – perhaps something like "(compared with the other simulations in Table 1)"

Corrected

240: "day of the week its emissions…" – "day fo the week on which its emissions were injected"

Restated as "Our "Smoke Age" simulation (see Table 1) has the brown carbon tracer "tagged" in such a way as to determine the day of the week on which its emissions were injected."

243: It's too late to change this study now, but surely using a non-perishable tracer such as Julian day would be better?

We will consider this in future work.

265: "closer together is to the translate" – delete "the"

275: "After the Colarco et al study…" – "Since the study by Colarco et al…"

282: "almost the same" – how do you quantify this?

283-4: "See Colarco et al for a detailed description of the AI simulator" – no need to say the same thing twice…

Here and previous three comments (highlighted), we have simplified the write up in this section so it now reads: "We employ a radiative transfer code to simulate the OMI aerosol index (Buchard et al., 2015; Colarco et al., 2017). Similar to the computation of the model AOD, the AI simulator takes as input the GEOS-simulated aerosol mass distributions and meteorological fields, and—

subjected to the GEOS aerosol optical property assumptions—simulated OMI radiances are calculated at the OMI viewing conditions (viewing geometry, terrain height, and surface reflectance). AI is then calculated as in Colarco et al. (2017). Because we do not explicitly simulate the impact of modeled cloud fields on the simulated radiance we restrict our comparisons to the highest quality OMI retrievals (formally, QA-flag = 0) to eliminate as much as possible cloud pixels in the satellite product impacting our comparisons."

286: "Linux-distribution" – is the hyphen necessary/correct? Your editor may have an opinion. 286-290: Three sentences in a row start with "We use…"

293: It's not clear how these offsets are done. Is the entire meteorological field shifted while keeping the initial position the same? Is there some reference describing this in detail?

294: "all-possible" – hyphenation is dubious.

Here and two previous comments we have rewritten this section for clarity: "We use the Linux-based distribution of the NOAA (National Oceanic and Atmospheric Administration) Hybrid Single-Particle Lagrangian Integrated Trajectory (HYSPLIT, v5.2.1) model (Rolph et al., 2017; Stein et al., 2015) to understand the transport pathway and origin source locations of the smoke observed during ORACLES 2016. \ERA-5 meteorological data is used to drive HYSPLIT so that the trajectories calculated are consistent with our GEOS simulations. We employ the meteorological grid ensemble approach within HYSPLIT to quantify the uncertainty and divergence associated with the trajectory calculations. In this method, trajectories are computed for a 3-dimensional cube centered on the starting point. Instead of moving the trajectory initial location about the starting point, however, all the trajectories start from the initial point but the driving meteorological data for each trajectory is offset slightly from that central location (see https://www.ready.noaa.gov/documents/Tutorial/html/traj_ensem.html, last accessed: 16 October, 2023). Default offsets are chosen, so that the meteorology is spread one grid box in the horizontal (~25 km) and about 250 m in the vertical. This results in 27 members of the trajectory ensemble for all possible offsets in X, Y, and Z directions in space.

299: "Sept 2016 in context of…." – "in the context of…"

Corrected

302: The aircraft did not observe anything, they just fly through the air. The instruments do the observing (or perhaps strictly speaking, not even that; however this is more a question of philosophy) Perhaps: "…observed from both the P-3 and ER-2…"

Changed to "observed from"

307: "We suggest the vertical variation of SSA…." This seems completely out of context here. Is this one of your hypotheses?

Sentence is removed.

310: "vertical flight trajectory" – "vertical flight profile". "Trajectory" is strictly speaking something that is followed by e.g. artillery shells or re-entry vehicles. In the horizontal plane, the aircraft would follow a "track".

Changed the word to "profile"

311: What is the time difference between the airborne measurement and the GEOS model time-step?

The GEOS model internal time step is 7.5 minutes. The observations are available every 1 second on the ORACLES merged data files.

313: "... with part of P-3 flight path" – "with part of the P-3 flight track".

Corrected

316 & Fig 3 & Fig 4. The P-3 appears to have been flying northbound until approx. 11:30UTC, then turning southbound. The ER-2 appears to have been flying northbound until around 10.1 UTC. Thus, these plots are folded back on themselves in latitude. This is quite confusing, and a clearer description of these plots is warranted.

We attempted to clarify the description: "On September 24 the P-3 flew an out-and-back south-to-north-to-south flight along 11°E longitude to and from Walvis Bay, Namibia (Fig. 1b). The P-3 vertical flight profile is shown in Fig. 3 with the in-situ (PSAP and nephelometer) measured dry extinction superimposed on the baseline GEOS-simulated extinction profile sampled in space and time along the flight track. The out-and-back nature of the flight track is evident in the model fields, which show a quasi-symmetric vertical profile in time. The blue stars on Fig. 3 indicate the location of 4STAR sky-scans for which quality screened column-integrated SSA were retrieved (Pistone et al., 2019). On the same day the ER-2 aircraft carrying the HSRL-2 lidar spatially overlapped geographically with part of the P-3 flight path (Fig. 1b). The retrieved and GEOS-simulated vertical aerosol extinction profile (GEOS sampled along the ER-2 track) are shown in Fig. 4. A similar aerosol plume structure is apparent in GEOS comparisons to these two sets of observations (Figs. 3 and 4)."

317: Refer to the specific plot (i.e. Fig 3?). "About 13UTC" is very imprecise; the profile starts well before 13UTC. Please furnish the exact times that you are referring to. Similarly: "About 6km to 1km" – please furnish the exact altitudes that you are referring to. Also, improve the horizontal scale markings on the plot so that the reader can follow these exact times that you furnish.

Clarified to: "Between 1245 and 1301 UTC the P-3 sampled a multi-layered smoke plume around 12.3°S and 11°E while descending from 6 km to 1 km."

318: (~9.5 UTC) – please furnish the exact time. You highlight a box prior to 9.5 UTC on the plot. Is this the time you refer to, or is it the high-extinction plume between 3.2km and 5.2km altitude, shortly after this time that you wish to draw the reader's attention to?

Attempted to clarify the discussion: "Between 1245 and 1301 UTC the P-3 sampled a multi-layered smoke plume around 12.3°S and 11°E while descending from 6 km to 1 km. The ER-2 flew over the same location earlier in the day, and this common spatial region is shown by the black rectangular box in Figs. 3 and 4 and is indicated by the green star in Fig. 1b. For purposes of comparison, we averaged the ER-2 HSRL profiles over the same area covered during the P-3 profile (from 915 to 930 UTC, black box in Figure 4). We compared the observed aerosol extinctions for this profile based on in-situ instruments (Fig. 5a) and HSRL-2 lidar (Fig. 5b) with our GEOS baseline simulation. The in-situ observations are made under dry conditions (RH<=40%), while the lidar observations are at ambient conditions. An elevated smoke layer was observed from both aircraft between 4 – 6 km altitude and a lower layer between about 1.5 – 3.5 km altitude."

322: "extinction magnitudes can be explained in part..." – This is a rather qualitative assessment, with a equally vague description. Can you provide a quantitative assessment of this?

Clarified as: "The model profile of humidity agrees well with the observations (Fig. 5c), so the difference in extinction suggests that the model has more hygroscopic growth of the smoke at for the higher altitude plume than the observations suggest, a point we will return to later."

325: "at least for the upper level smoke layer" – "upper level smoke layer" might be subject to interpretation. Starting off your analysis of the profiles with a precise description of what you see (e.g. an upper smoke layer measured by the ??? instrument in the P-3 aircraft while descending between 4.2 – 5.2 km and a lower smoke layer at 1.6- 3.6km...) would get you and your reader on the same page and allow you to clarify a lot of the vague language in these and following paragraphs.

We hope the responses to the previous two points have clarified the concern here.

325: "model simulated" – your editor might want a hyphen here, since "model-simulated" is an adjective, although Copernicus language editors seem to ignore this common feature of English grammar.

Corrected.

Fig 5c: have you tried plotting temperature on this plot? The existence of absolutely stable layers in the winter atmosphere over the sub-continent is a feature that was described decades ago by Tyson. It would be interesting to see that they exist over the ocean too.

We have plotted the temperature profile on Figure 5c and added the following text: "An elevated smoke layer was observed from both aircraft between 4 – 6 km altitude and a lower layer

between about 1.5 – 3.5 km altitude. Fig. 5c shows the P-3 measured temperature for this profile, as well as the dry and moist adiabats anchored at 2 km. The slope of the observed temperature profile is greater than the dry adiabatic lapse rate but less than the moist adiabatic lapse rate, indicating conditionally stable air layers that explain the distinct plumes over the ocean, also a feature over land (Tyson et al. 1996)."

326: "having a very good match of simulated RH" – delete "having"

Corrected.

339: "underestimated in the model overall" – by what measure? can you quantify this?

339-340: "multiplume" – this is not an established term, and has all sorts of connotations which will differ from one reader to another. Perhaps "layered" is better?

341: "Here, BC is the primary…" are you referring to a specific figure, or a specific profile, or a specific model simulation? This is not clear.

Here and previous two comments, we have restated the text as: "OA is overestimated by the model for altitudes below 2.5 km (Fig. 6a), while nitrates are underestimated for the same altitudes (Fig. 6c). Sulfate (Fig. 6d) and BC (Fig. 6b) show a similar profile to the simulated OA but are present at lower concentrations than the observations suggest, particularly for the lower altitude layer. Sulfate is about half the concentration in the model as observed. BC has similar concentration to the observations for the higher smoke plume (~1 µg m$^{-3}$) but is only about half the concentration of the observations in the lower layer. With the exception of nitrate the observed multi-layer structure for this profile is evident in the model species."

343: Same comment. Are you referring to a specific figure?

Added reference to Fig. 5d

346: "(Table 1, Section 2.4.1)" – it would be polite so say "(described in Table 1 and Section 2.4.1)" rather than barking at your reader.

Corrected as suggested.

346: "almost monotonic" is meaningless. It's either monotonic or not, and it seems from the plot that is is clearly not. What do you mean?

351: "demonstrating that the models are in close agreement" – by what metric?

Here and previous comment, we have restated the presentation as: "Figure 7a also shows the smoke age derived using the WRF-AAM (Weather Research and Aerosol Aware Microphysics) model (Saide et al., 2016) that was used for forecasting and flight planning during the ORACLES

campaign (Redemann et al., 2021). Both GEOS and WRF-AAM models have a similar overall structure of simulated plume age, showing minimal smoke age around the center altitude of the upper smoke plume (~ 4 days at 4.5 – 5 km altitude) and higher smoke age at lower altitudes, with a local maximum of about 5 – 6 days between 3 – 4 km and increasing to 7 – 8 days below 2 km."

352 & Fig 7b. Would it not be clearer to use something like a box-and-whisker plot on the vertical profile. How does the distribution of extinction fraction in the "clean" layer at 3.8km look?

Thanks for the suggestion, but we have left Figure 7b as initially presented. We feel the histogram style used here also adequately conveys the information we are trying to communicate. Looking at the extinction profile in Fig 7a, the layer at 3.8 km is not "clean", but just lower in extinction magnitudes compared to the smoke layers above and below it (~ 100 Mm$^{-1}$ versus ~ 160 Mm$^{-1}$). Either way, we did try and plot the smoke age distribution at 3.8 km, and we found the age distribution at this height to be intermediate between the 3 and 4.5 km curves. Therefore, to avoid clutter, we do not present it here.

359: "…causing the weighted mean smoke age to be younger than it possibly is" could this be rephrased to make it clearer?

We have rephrased the text: "Finally, for GEOS, we are restricted by the way we track the smoke age that we can only resolve smoke age up to 7 days, and older smoke is lumped into this last bin of our histogram so that the effective age computed is slightly younger than if we could account for all possible smoke ages."

362: "… composition of aerosols change with smoke age" – "changes"

Corrected

366: If these quantities are correlated, could they be plotted against each other? This would make your point clear, rather than the qualitative "by inspection" correlation that you claim.

We have added panels to Figure 8 that show the correlation of OA:BC ratio with SSA (e) and NO$_3$:BC ratio with SSA.

369: "nitrate:BC and SO4:BC" – consistent nomenclature: choose words or formulae

Rewritten as: "NO$_3$:BC and SO$_4$:BC"

374: This introduces your hypotheses, however this sentence is quite weak, almost an afterthought.

We rephrase as: "The model's failure to simulate the observed OA:BC ratio variability with age is correlated its failure to simulate the observed variability in SSA. In the following we consider two

hypotheses to explain the age-related variation in the OA:BC ratio. In the first we consider the possibility that smoke of different ages may be originating from different source regions with different emissions of OA and BC. In the second we consider the possibility that there is some unsimulated mechanism for the loss of OA during transport that is related to its age."

377: "with perhaps different..." – this is your hypothesis. "Perhaps" makes it weak and negotiable. State it clearly and boldly.

Restated as: "The first hypothesis we examine is whether smoke of different ages is originating from different source regions with different characteristics in the composition of emitted species or different proportions of flaming to smoldering phase of the combustion products. "

380: "origin locations" – delete locations.

Corrected.

385: It's not clear how Fig 9a is arrived at. Is this from QFED introduced in line 216, then never mentioned again?

We have added clarifying text: "GEOS uses biomass burning emissions from the QFED inventory, based on the MODIS fire radiative power products (Section 2.4). QFED provides daily, gridded biomass burning emission fluxes of relevant species, such as OA, BC, and $SO_2$. Figure 9a shows the QFED emission locations and cumulative amounts of OA emitted over the seven days prior to September 24."

390-392: "further suggest that the contribution... are possibly causing the..." – "suggest" + "possibly" in one sentence makes for overwhelming uncertainty. What do you want to say here?

393: "suggest... almost..." here makes me think that you have no faith in this tool whatsoever. Why do you use it?

Here and previous comment, we have rewritten: "For the higher-altitude smoke layer (centered around 4.5 km), the clustering of the trajectories and their intersections with the surface (where they would entrain smoke) occurs only a few days before intercepting our profile location (the black star in Fig. 9b, at 12ºS, 11ºE), consistent with the smoke being young (about 4 days old, Fig. 7). By contrast, the lower-altitude initialized back trajectories (originating at 2 km, Fig. 9c) intercept the surface several days back and further to the east of the profile location, consistent with the smoke at these levels being older (about 6-7 days old, Figure 7)."

396: "as a proxy for vegetation type". Surely the OA:BC is *determined* by vegetation type?

We have added some clarifying text: "The trajectory information alone does not tell us that the composition of the smoke in the two different layers is similar. Our "Smoke Composition" simulation (see Table 1 in Section 2.4) is used to distinguish the contributions of individual

vegetation types to the total smoke composition. QFED distinguishes among several different vegetation types according to land use datasets, and vegetation-dependent emission factors are used to scale from biomass burned to emissions of specific species (i.e., OA and BC). In the "Smoke Composition" simulation we "tag" the smoke emissions from each vegetation type so we can track its evolution separately."

399: Fig 10 is introduced rather suddenly here, and the reader is left to figure out its meaning with no guidance from the author.

We have added: "In Figure 10a we quantify the contributions of emissions from individual vegetation types to the smoke composition at our profile location, separately for altitudes of 4.5 and 2 km, the central altitudes of the two smoke plumes in the profile."

407: "do not even differentiate" – delete "even"

Corrected

409: "the most different fuel types" – it's not clear what is meant by this.

We have clarified the text: "After savannas and grasslands the next most prevalent vegetation type contributing emissions is forest, which has a higher OA:BC emission ratio than grasslands and savannas, but contributes only about 10% to the total smoke load and is similar in contribution for both plumes. Crop and agricultural residue has distinct OA:BC ratios (that are highly uncertain, see Table 2) but are an even smaller contribution to the total aerosol load and are also a similar contribution to each layer."

410: savannas: check for consistent spelling cf. line 422.

We have adopted the spelling of "savanna" throughout.

Table 2: where does the multiplier of 1.8 come from? Is this an empirically established ratio? Is there a reference for this?

Clarifying text has been added to the caption for Table 2: "Also shown is the OA:BC ratio used in the GEOS simulations. GEOS assumes OA:OC ratio of 1.8, based on airborne mass spectrometry measurements (Hodzic et al. 2020)"

417: "emission ratios based to fire characteristics…" – "based on fire char…"

Corrected

422: spelling of savannah, cf. line 410.

We have adopted the spelling of "savanna" throughout.

422: "savannah and grassland region" – plural "regions"

Corrected

424: "outside plume (>6km) values" – "values outside the plume (found at altitudes > 6km)"

Corrected

427: "Therefore..." here you refute your hypothesis. I would consider having this sentence stand as a paragraph on its own.

We have rephrased as: "Based on the analyses shown here we conclude it is unlikely that the observed differences in smoke composition (that is, the OA:BC ratio, as in Figure 8b) at different vertical levels are due to differences in either the burning source vegetation type or combustion conditions."

442: "increase in particulate oxygen" – it's not at all clear what this is. Is this a change in oxidation state? Are there unbound oxygen atoms resulting from a chemical reaction? Are particles being oxidized?

We have rephrased the text to hopefully increase clarity: "In the gas phase, two main processes can affect the volatilities of organics during atmospheric oxidation: fragmentation and functionalization. Fragmentation refers to the loss of carbon from the organic particles, whereas functionalization refers to an increase in particulate oxygen due to the addition of polar functional groups. Therefore, fragmentation leads to an increase in vapor pressure (i.e., the organic compounds become more volatile), while functionalization leads to lowering of vapor pressure of the organic compounds. In the condensed phase, additional bimolecular processes, such as accretion/oligomerization reactions can also affect volatility (Kroll et al., 2009)." The following figure from Kroll et al. (2009) explains the point:

[Figure]

**Fig. 1** Example of fragmentation and functionalization pathways in the atmospheric oxidation of an organic compound. In the oxidation of particulate organics, fragmentation leads to a loss of carbon from the particle (assuming at least one of the fragments is volatile), whereas functionalization leads to an increase in particulate oxygen.

448: "prescribed at emissions…" – at the point of emission?

Corrected to "at the point of emission"

448: "the burning fuel type" – delete "burning"

Corrected

452: "further aged smoke" – in plain language: "older smoke"

Corrected

462: "brought the modeled SSA curve closer to 4STAR" – by what metric? Some wavelengths look "close", some wavelengths look "far". What are the implications of this? If you are giving a qualitative assessment, base it on some numbers.

We have modified the text: "Overall there is better agreement in the SSA between 4STAR and the OA-loss simulation than 4STAR and the Baseline, except at the shortest wavelength (Fig. 11c, where at 400 nm we have $SSA_{4STAR}$ = 0.88, $SSA_{Baseline}$ = 0.87, $SSA_{OA-loss}$ = 0.86 and at longer wavelengths the $SSA_{OA-loss}$ is closer to 4STAR than the Baseline by a magnitude of 0.02; see also Table 3)."

*Table 3. Comparison of the observed and modeled mean SSA between 4STAR and GEOS experiments at 400, 500, and 675 nm.*

| Wavelength (nm) | Mean SSA | | | |
|---|---|---|---|---|
| | 4STAR observations | GEOS baseline | GEOS OA-loss | GEOS OA-loss+updated optics |
| 400 | 0.88 | 0.87 | 0.86 | 0.87 |
| 500 | 0.86 | 0.90 | 0.88 | 0.88 |
| 675 | 0.84 | 0.89 | 0.87 | 0.85 |

470: you could refer to Fig 10 here, if I understand it correctly.

Reference is made to Figure 6

485 & Fig 13: f(RH) nomenclature in Fig 13 axis labels is confusing and unclear.

We have simplified the labeling on Fig. 13

486: f(RH) is high/low, whereas here you mention low followed by high. This is confusing.

Rewritten as: "We consider 80% and 10% as the high and low RH, respectively…"

489: here you use measurements between 1.5 – 6km. On line 479 you use 1.0 – 5km ostensibly to achieve the same aim.

The two different altitude ranges are referring to two different analyses, one for particle size and the other for hygroscopicity.

496: "Sulfate hygroscopicity increases..." do you mean "hygroscopic growth factor"? These are not the same thing.

Corrected to: "Sulfate hygroscopic growth factor"

501: "tracks with BC" – this is an extreme colloquialism. Meaningless. What are you trying to say?

We have clarified: "Finally, we consider that the hygroscopic growth of OA is the same as for BC, and here we see the closest match to observations (a reduction of the simulated f(RH) from greater than 2 to about 1.7, compared to the value of 1.4 determined from the observations, see Fig. 13a, and also the reduced ratio of the simulated-to-observed f(RH)), and so this comprises our second adjustment to model assumption of OA properties."

501-2: "closest match to observations" - can you provide some numbers that you used to guide your assessment?

See response to previous comment.

506: "blue (or 355nm) channel" - is it blue, or is it 355nm? or is it blue and 355nm?

Clarified as 355 nm

507: "by (Veleovskii et al, 2020) : V. et al (2020)

Corrected

510: "were set" – past tense? "was set..."

Corrected

511: Give the precise definition and be done with it. If there is a need for a looser definition, provide this afterwards.

We have simplified the text, writing: "This change was motivated by our finding that our model-derived lidar ratios (that is, the ratio of extinction to backscatter)"

Fig 14 & lines 510, 517, 519, 520: is it k_340 or k_350? The figure and the text do not correspond.

We have corrected this. It is k_350 throughout.

521: Why is it worth mentioning this? How does this impact your study? Perhaps a better phrasing would be "it is worth mentioning that GOCART assumes particles to be spherical, notwithstanding Meng et al (2010)..."

We have removed this text.

525: "we call as 'updated optics'..." – delete "as"

Corrected

527: "these optics update" – "this optics update"

Corrected

528: "slight decrease" – what constitutes a slight decrease?

529: "very close" – what constitutes very close?

For this and the previous comment we have restated: "For the in-situ profile at 550 nm, there is a small decrease in model SSA with "OA-loss + updated optics" compared to the "OA-loss" case (< 0.01), and it remains within 0.02 of the observed value at all altitudes (Fig. 11b)."

533: "assumptions that is leading to.." – "that are leading to"

Corrected

534: Explian by what metric it shows a "better match"? You have some measurements here that took incredible resources to make, yet you provide a qualitative description.

We have added Table 3 above and modified the text here: "However, overall, the "OA-loss+updated optics" case shows the best agreement with the 4STAR observations compared to the other two cases for wavelengths less than 700 nm (Table 3). At longer wavelengths the improvement is less clear; although "OA-loss + updated optics" is closest to the 4STAR observations at 870 nm of all our experiment, it is the worst agreement at 1000 nm. We did not investigate other aerosol components (e.g., dust) that could contribute to especially longer wavelength impacts."

541: "the September" – delete "the"

Corrected

543: does not deviate much. Add some numbers to your qualitative assessment. These measurements took incredible resources to make.

We have rephrased the text: "The model shows a very good agreement with AERONET observations over Lubango in terms of both AOD and SSA, and the final "OA-loss + updated optics" case has overall similar performance to the model baseline case (Fig. 15a, c) and improves the SSA simulation at 440 nm and 675 nm (Table 4)."

And added Table 4:

*Table 4. Comparison of the AERONET and GEOS SSA at two AERONET locations during September 2016.*

| Wavelength (nm) | Location | Monthly Mean SSA | | |
|---|---|---|---|---|
| | | AERONET observations | GEOS baseline | GEOS OA-loss+updated optics |
| 440 | Lubango | 0.88 | 0.87 | 0.88 |
| | Mongu_Inn | 0.88 | 0.84 | 0.87 |
| 675 | Lubango | 0.87 | 0.88 | 0.86 |
| | Mongu_Inn | 0.86 | 0.85 | 0.84 |

546-7: "suggests that model is missing" – insert "the"

Corrected.

552: "over complete SE Atlantic" – "the broader SE Atlantic region" or something like that.

Corrected as suggested.

556: "the Model and MODIS NNR AOD" – it would be worthwhile to remind the reader here how the model AOD is derived.

We added the following text to Section 2.4 to explain how this is done: "Optical properties are defined in pre-computed lookup tables that are a function of species, wavelength, and relative humidity (to account for particle humidification). In-situ optical quantities such as extinction and SSA are computed by summing across the aerosol species concentrations as scaled by the appropriate optical property (e.g., mass extinction efficiency), Column integrated optical quantities (e.g., AOD and SSA) are computed as the vertical integral of the in-situ properties."

557: "Overall, there is a good match..." – by what metric?

We have redrawn figure 16 and clarified the text: "The last column (Fig. 16c&e) shows the differences between the model and the observations. Broadly, as depicted in the AOD difference plots, both model simulations show a good agreement with the NNR retrievals, especially over the ocean and the smoke outflow region compared to the source region over the continent."

558: "...compared to the source region..." it's not clear what this means. Is the "overall good match" better over the ocean than over the source region? By what metric?

See previous response, and also: "In our final simulation, however, this high bias in model AOD is reduced from +0.02 to -0.02 (Fig. 16d&e). Closer to the burning sources over land, this decrease in model AOD in our final simulation is due to the adjusted hygroscopic growth for the OA particles (see section 3.2), whereas over the ocean, as the plumes move away from the continental burning sources, the decrease in model AOD is due to the accounting of OA loss with increasing smoke age. Finally, we also present the performance metrics (mean bias, root-mean-square error or RMSE, and Pearson correlation coefficient or r) to quantify the agreement between the model simulations and the observations with a focus over the ORACLES-I region (Fig. 16c&e). The statistical comparisons further emphasize that inclusion of OA loss processes and adjustments to model OA optics does not deteriorate the model performance in terms of AOD simulation, instead makes it marginally better compared to baseline run by lowering the RMSE (from 0.11 to 0.08) and increasing the r values (from 0.91 to 0.93)."

Perhaps a plot of AOD from MODIS and the model runs along the 10deg S latitude line would illustrate the point you are trying to make more clearly.

Thank you for the suggestion, but we think now, with the addition of performance metrics to Fig.16 and revision of the text within this section, the point we were trying to make is clearer and does not require further emphasis or plotting of AOD along 10deg S.

575: you speak of retaining OMI pixels without cloud contamination, in the context of the model-based AI calculator. This is clearly meaningless; the discussion about cloud contamination should come after you introduce the OMI observations in line 576.

We clarify the presentation this way: "In Fig. 17 we show the GEOS-simulated OMI aerosol index (AI) for September 2016 in comparison to the OMI retrievals. The OMI retrieved AI is shown in Figure 17a. To minimize the impact of clouds on the comparison we retain only OMI pixels with QA=0 (low probability of cloud contamination). For the model-calculated AI, the GEOS-simulated aerosol profiles are sampled at the OMI footprints. The model optical property assumptions are applied to the simulated aerosol profiles, and with the OMI observation geometry and retrieved surface reflectance are input to the AI calculator, which simulates the OMI radiances. The simulated radiances only include terms for the aerosol, molecular background, and surface."

585: I can see hardly any difference between Fig 17c and Fig 17e. How do you quantify the "much more favorable comparison"?

There was an error in the original submitted figure where the same figures were intended to refer to different experiments. We have redrawn Figure 17 and simplified the presentation. Additionally, we add some quantitative discussion: "The apparent discontinuity in the AI magnitude between land and sea along the west coast of southern Africa is a sampling artifact brought on by cloud screening, with nearly 3x as many OMI pixels retained over land as for over the coastal ocean. For this reason, we restrict a quantitative assessment of the model performance to the boxed region over land shown in Fig. 17c&e, and statistics are reported in the figure panels. In the indicated region there is an improvement in the bias of the modeled AI from +0.28 to -0.14 and a reduction in the RMSE from 0.42 to 0.30 in moving to the updated aerosol optics."

591: The direct radiative forcing of the aerosols is what it is in the real world out there over the SE Atlantic. You are talking about the modeled DRF.

We refer now to the "modeled direct radiative forcing".

598-600: "due to aerosol difference between our two runs... the other without the aerosols" – do you mean without aerosols, or do you mean the difference between the baseline model and the modified/updated aerosol scheme?

We have added some clarifying text: "For each model run, the radiative forcing is computed by two successive calls to the radiative transfer code, one including the effects of aerosols and the other without the aerosols."

603: this is the first mention in this paper of the "main cloud feature". What is this cloud feature?

We add some earlier mention of the cloud feature now: "Clouds are not much changed between the two model runs, so in Fig. 18b we show the low cloud fraction from the baseline run, which shows a main cloud feature (cloud fraction > 40%) centered at 10º – 15ºN extending west of the continent."

606: "reflected radiation to space" – "radiation reflected to space"

Corrected

607: it's not clear where the border of DRC and Zambia is. What lon/lat are you referring to.

We have clarified the text: "The broad positive SW forcing difference over the continent reflects the relatively greater backscattered solar radiation over the dark continental surface corresponding to the higher AOD in that region in the updated model run (region of over-land negative AOD difference in Fig. 18a)."

613: You introduce the cloud features in the last line of your section. What clouds are you referring to? Modeling of cloud features has not been mentioned prior to this.

Explanatory text added earlier addresses the cloud features.

621: "move observing smoke plume" – do you mean "absorbing"?

Corrected

622: "We hypothesize a loss process … explains…" - "We hypothesize a loss process … that explains" or "loss process… explaining"

Corrected

624: "mimic" is again an odd anthropomorphizing of the model.

Corrected

634: "a better performance…" issues of how you quantify this have been addressed in the sections above.

639: consider deleting "In terms of future directions" and start the sentence with "The simplistic approach…"

Corrected as suggested.

643: "mimic"

Corrected.

Reviewer #2

This work evaluated NASA GEOS model's ability to simulate biomass aerosol properties by comparing with observations from field campaign ORACLELS-I. Sampa Das et al. implemented two adjustments in GEOS model to improve organic aerosol aging process and optics caculation. These two adjustments were further evaluated by AERONET and satellite observations. Radiative implication of these adjustments were discussed in the end. The content aligns well with the ATMOS CHEM PHYS's scope and I recommend considering publication after major revisions.
Major comments:

- The abstract is almost identical with the first paragraph of conclusion. Abstract can be less detailed in methods, while the conclusion can provide more information about the adjustments implemented in the model.

We have adjusted the abstract as suggested.

- There are substantial text, figure and citation format issues in the draft. Details will be listed in the Minor comments section.

We have addressed these issues below.

- For the first adjustment, are the 60% increased OA and 15% increased BC emission consistent with previous studies, or just to compensate the OA loss process introduced in this work? For the second adjustment about hygroscopic growth, it is not too clear and please add more descriptions about this adjustment.

Our adjustments were practical, in part to compensate for the OA loss mechanism added, as noted by the reviewer, and also to compensate for too-low BC amount in the Baseline simulation (see Figure 6a). The absolute emission of biomass burning aerosols is subject to great uncertainty and is often tuned in models to achieve some objective (e.g., column AOD that agrees with observations). See, for example, Figure 2 in Pan et al. (2020) that shows that there is a greater than factor three range in organic carbon emissions across several commonly used databases.

**OC biomass burning emission for 2008**

**Figure 2.** The spatial distribution of annual total organic carbon (OC) biomass burning emissions for 2008 estimated by six biomass burning emission datasets (units: $\mathrm{g\,m^{-2}\,yr^{-1}}$). The global annual total amount for each dataset in 2008 is indicated in the parentheses.

Pan, X., Ichoku, C., Chin, M., Bian, H., Darmenov, A., Colarco, P., Ellison, L., Kucsera, T., da Silva, A., Wang, J., Oda, T., and Cui, G.: Six global biomass burning emission datasets: intercomparison and application in one global aerosol model, Atmos. Chem. Phys., 20, 969–994, https://doi.org/10.5194/acp-20-969-2020, 2020.

Regarding the humidification, we added some further explanatory text around Figure 13: ". Finally, we consider that the hygroscopic growth of OA is the same as for BC, and here we see the closest match to observations (a reduction of the simulated f(RH) from greater than 2 to about 1.7, compared to the value of 1.4 determined from the observations, see Fig. 13a and also the reduced ratio of the simulated-to-observed f(RH)), and so this comprises our second adjustment to model assumption of OA properties."

- From the spatial distribution differences in figure 16 and 17, it is not clear whether the models with the implemented adjustments are more consistent with the observations or

not. Please better quantify the differences by some metrics. Either regional mean differences or spatial correlation will work.

We have included regional statistics and some discussion about the improvements in the updated model versus the baseline. For Figure 16 we add: "In our final simulation, however, this high bias in model AOD is reduced from +0.02 to -0.02 (Fig. 16d&e). Closer to the burning sources over land, this decrease in model AOD in our final simulation is due to the adjusted hygroscopic growth for the OA particles (see section 3.2), whereas over the ocean, as the plumes move away from the continental burning sources, the decrease in model AOD is due to the accounting of OA loss with increasing smoke age. Finally, we also present the performance metrics (mean bias, root-mean-square error or RMSE, and Pearson correlation coefficient or r) to quantify the agreement between the model simulations and the observations with a focus over the ORACLES-I region (Fig. 16c&e). The statistical comparisons further emphasize that inclusion of OA loss processes and adjustments to model OA optics does not deteriorate the model performance in terms of AOD simulation, instead makes it marginally better compared to baseline run by lowering the RMSE (from 0.11 to 0.08) and increasing the r values (from 0.91 to 0.93)."
And for Figure 17 we have: "In the indicated region there is an improvement in the bias of the modeled AI from +0.28 to -0.14 and a reduction in the RMSE from 0.42 to 0.30 in moving to the updated aerosol optics."

Minor comments
For the improper text formats:
- Certain acronyms are repeated or mentioned more than once, such as AOD in line 175 and line 105, TOA in line 37 , line 178 and line 598. Please check other acronyms throughout the papers.

We have attempted to clean up the acronym definitions throughout the text.

- There should be spaces between the math operation symbols and the numbers, such as line 100: <40%, should be < 40%. Also the equal operations in line 629 and line 632. Please check this issue throughout the papers as well.

We have attempted to clean up the math operator spacing throughout the text.

For the improper citation formats:
- Citations for data sets including links should be corrected: line 97 and line 253.

We have tested the links embedded in the text and they appear to work.

- Citations formats when abbreviations are included are wrong: line 119 for HSRL-2, line 156 for OMI.

Corrected

- In-text                citation                formats                are                incorrect. –Line 108: Dubovik and King, (2000), no comma in the middle. –Line 214: Colarco et al., (2014), no comma in the middle.

Corrected

For the improper figure formats:
1
- Figure title formats are inconsistent between the figures, neither the font nor the sizes. Please keep the title formats consistent of all the figures.
- The units in the axis caption should not be italic, such as Mm−1 in figure 4, should in M m−1. Please check the unit formats in other figures and change properly.
- The titles of some subplots in the same figure are the same and unnecessary, such as figure 6(a)-(d), figure 9(b)-(c), figure 10(b)-(c), figure 11(a)-(b). You can either delete the title, or use other more informative texts.
- Subplots in figure 15 have the same legend captions. If the legends are the same for all the subplots, no need to show them in all subplots.
- The label notation (a)(b)(c)(d) is very inconsistent between the figures, sometimes outside the figure, such as figure 1, sometimes inside the figure, such as figure 4. Also, the labels sometimes are not clear if overlapping over other colors, such as figure 16. Please try to keep the position of these labels consistent among the figures and keep them clear.
- The names of the cases are inconsistent between the figures. For the OA loss adjustment cases, the notation is +OAloss 6days in figure 11, while it is GEOS + OAloss in figure 14 and GEOS OA-loss in figure 16. It is better to describe all the simulations in table 1. Create the cases names that are easier to put in the figures and be consistent among the figures.
- The legends in figure 13(b) are inconsistent with the text notations along the lines. Since they describe the same thing, you can delete either one.

We hope that we have addressed the reviewers concerns in the revised figures. We have adopted "GEOS OA-loss" consistently where appropriate.

---

## Referee Report (RR1)

Minor corrections for

egusphere-2023-1311:

S. Das et al: Improved Simulations of Biomass Burning Aerosol Optical Properties and Lifetimes in the NASA GEOS Model during the ORACLES-1 Campaign

17 Feb 2024

The following refer to lines in the Author's tracked changes document:

201: "wind-blown source function" – this is not clear and has connotations of sea-salt aerosol generation by wind-blown whitecaps leading to increased AOD (cf. e.g. Smirnov et al 2012 https://doi.org/10.5194/amt-5-377-2012), or bubble bursting (cf. e.g. Prather et al 2013 https://doi.org/10.1073/pnas.1300262110). DMS is not generated by the action of the wind. Lana et al 2011 refer to the rate of transfer of DMS dissolved in seawater to the atmosphere being proportional to wind-speed. Is this what is implemented in the model?

274: "possible cloud pixels in the satellite" : "cloudy"

305: "spatially overlapped geographically" : "spatially" & "geographically" mean basically the same thing.

345: "… similar profile shape…."

375: "strongly but negatively correlated": e.g. "show a strong negative correlation"

519: "Figure 13a showed…" – rather stick to the present tense.

575: "have a systematic low bias compared to the AERONET observed AOD" – suggest: "systematic low AOD bias compared to AERONET observations"

575: "but nonetheless can capture its daily variability" – this suggests intra-day variability. Perhaps "day-to-day" is better.

578: "…case here corrected for the excessive absorption…" it's not clear what is meant. Is the AERONET data corrected by the authors, or did the "OA-loss+updated optics" treatment correct the bias that is observed relative to AERONET?

---

## Author Response (AR2)

**Author's Response on "Improved Simulations of Biomass Burning Aerosol Optical Properties and Lifetimes in the NASA GEOS Model during the ORACLES-I Campaign" by Das et al.**

We thank the reviewers and the Editor for their careful review of our revised paper and acceptance of our manuscript.

1. We have now implemented most of the additional "suggested word usage" recommendations from reviewer#1 (or suggestions listed in report#2) in our final revised version.
2. We also appreciate the Editor's interest in our study and the interesting observation with regards to the AERONET comparisons at the Mongu site. We aim to explore it further in our consequent investigations.
3. We have also updated the Figure 5b caption to now include the description of gray shading that depicts the one standard deviation area around the mean extinction.